# *Homo sapiens* lithic technology and microlithization in the South Asian rainforest at Kitulgala Beli-lena (*c.* 45 – 8,000 years ago)

Andrea Picin[1,2]*, Oshan Wedage[1,3]*, James Blinkhorn[4,5], Noel Amano[1], Siran Deraniyagala[6†], Nicole Boivin[1,7,8,9], Patrick Roberts[1,7]*, Michael Petraglia[7,8,10]*

1 Department of Archaeology, Max Planck Institute for the Science of Human History, Jena, Germany, 2 Bereich für Ur- und Frühgeschichtliche Archäologie, Friedrich Schiller Universität Jena, Jena, Germany, 3 Department of History and Archaeology, University of Sri Jayewardenepura, Gangodawila, Nugegoda, Sri Lanka, 4 Pan-African Evolution Research Group, Max Planck Institute for the Science of Human History, Jena, Germany, 5 Department of Geography, Royal Holloway, University of London, London, United Kingdom, 6 Department of Archaeology, Government of Sri Lanka, Colombo, Sri Lanka, 7 School of Social Science, The University of Queensland, Brisbane, Australia, 8 Department of Anthropology, National Museum of Natural History, Smithsonian Institution, Washington, D.C., United States of America, 9 Department of Anthropology and Archaeology, University of Calgary, Calgary, Canada, 10 Australian Research Centre for Human Evolution, Griffith University, Brisbane, Australia

† Deceased.
* picin@shh.mpg.de (AP); wedage@shh.mpg.de (OW); roberts@shh.mpg.de (PR); m.petraglia@griffith.edu.au (MP)

**Data Availability Statement:** All relevant data are within the manuscript.

**Funding:** This research is supported by the Max Planck Society and the University of Sri

## Abstract

Recent archaeological investigations in Sri Lanka have reported evidence for the exploitation and settlement of tropical rainforests by *Homo sapiens* since *c.* 48,000 BP. Information on technological approaches used by human populations in rainforest habitats is restricted to two cave sites, Batadomba-lena and Fa-Hien Lena. Here, we provide detailed study of the lithic assemblages of Kitulgala Beli-lena, a recently excavated rockshelter preserving a sedimentary sequence from the Late Pleistocene to the Holocene. Our analysis indicates in situ lithic production and the recurrent use of the bipolar method for the production of microliths. Stone tool analyses demonstrate long-term technological stability from *c.* 45,000 to 8,000 years BP, a pattern documented in other rainforest locations. Foraging behaviour is characterised by the use of lithic bipolar by-products together with osseous projectile points for the consistent targeting of semi-arboreal/arboreal species, allowing for the widespread and recurrent settlement of the wet zone of Sri Lanka.

## Introduction

During the Late Pleistocene, hominins reached new geographical areas as they expanded out of Africa, populations eventually extending to the far western [1–5] and eastern edges of Eurasia [6–9]. As groups extended their range, they modified their toolkits to introduce novel technologies and subsistence strategies as they engaged with new climatic zones and terrestrial ecosystems [10–16]. In the last few decades, archaeologists have placed increasing emphasis on

Jayewardenepura. A. Picin is funded by the German Research Foundation (DFG – project STONE, n° 429271700), and collaborates in the Spanish MICINN project PID2019-103987GB-C31. The funders had no role in study design, data collection and analysis, decision to publish, or preparation of the manuscript.

**Competing interests:** The authors have declared that no competing interests exist.

the role that environmental variability played in influencing the emergence and evolution of *Homo sapiens* [17,18]. Our species appears to demonstrate increased levels of behavioural plasticity that enabled populations to adapt to extreme environments such as those of the arctic, high-altitude plateaus, and tropical rainforests [17,19,20]. There has been increasing archaeological interest in the technological and behavioural strategies employed by *Homo sapiens* in different parts of its increasingly global distribution between 100 and 45,000 years ago [9,21–24].

Traditionally, tropical rainforests were seen as ecological barriers for Pleistocene human occupation based upon the perceived scarcity of carbohydrate-rich plants and fat- and protein-rich fauna [25–27]. Colonization of these ecosystems was considered to only have been possible with specialized Holocene toolkits or cultivated resources [28,29]. Recent investigations in Sri Lanka (South Asia), however, have been part of a wider body of research that has challenged this hypothesis, providing concrete, multi-disciplinary evidence of specialized foraging and hunting activities from 48,000 years ago [30–34] along with early microlithic [35–40] and bone toolkits that have been hypothesized to relate to the development of projectile technologies to target arboreal prey [30,33,41]. These discoveries are important for understanding how technologies allowed early groups of *Homo sapiens* to exploit challenging habitats. Although tropical rainforests have a high primary biomass, that is less impacted by seasonality, the exploitation of their resources is not an easy task. Fruits are located high above the ground, and seeds often contain toxins and tough outer coats needing intense processing to become edible [42,43]. Moreover, dense forested vegetation makes the identification and attainment of possible prey difficult [42]. Late Pleistocene occupation of tropical rainforests would therefore likely have necessitated specialized subsistence strategies, planning, and honed toolkits.

In the past few decades, fieldwork in caves and rockshelters of Sri Lanka's Wet Zone has expanded our understanding about human settlement [35,39,44]. Three sites—Batadomba-lena, Fa-Hien Lena, and Kitulgala Beli-lena–preserve horizons from the Late Pleistocene to the Holocene [35,39] as well as the oldest human remains of *Homo sapiens* in South Asia [45,46]. Multidisciplinary studies on the preserved remains at these sites have revealed continuous exploitation of the rainforest supported by the hunting of arboreal and semi-arboreal faunal species [31–33] and the collecting of freshwater/terrestrial molluscs [34,37] and wild fruits (e.g. breadfruit, kekuna nut) [34,37,47]. Reassessment of cultural materials have demonstrated that technical behaviours were associated with the consistent production of bone points and quartz microliths (< 40 mm) [30,33,34,36,41,48]. Thus far, detailed descriptions have been reported for the lithic assemblages of Batadomba-lena [36,48] and Fa-Hien Lena [38]. However, comparable work on the site of Kitulgala Beli-lena has not been conducted. In order to develop a broad understanding of the adaptive strategies employed by early groups of *Homo sapiens* to the Sri Lanka's rainforest, here we undertake a comprehensive technological study of the lithic industries of Kitulgala Beli-lena. This information enables examination of the development of foraging in tropical environments and discussion of the behavioural complexity of the human toolkit evident since the onset of the settlement in the rainforests of Sri Lanka in the Late Pleistocene.

## Mobility and hunter-gatherer tool-kits

Ethnographic observations of hunter-gatherers point to a close relationship between subsistence strategies and inhabited ecosystems [42,49,50]. In his seminal paper, Binford [51] formulated the *forager-collector* continuum as a theoretical framework for estimating the responses of hunter-gatherers to the distribution of resources (e.g. food, water, fuel) across the landscape

and through time. In areas where resources are scattered, hunter-gatherers tend to move their residential camps frequently. In contrast, in a high biomass environment, hunter-gatherers often settle in foraging zones where resource accessibility allows for frequent logistical forays and few residential moves per year [42,49,51]. These different types of land use patterns are reflected in various levels of investment and planning in technological organization, explored by Binford [52] through the concept of curation. Curated technologies are characterized by a high-level of investment in manufacture and maintenance. Conversely, expedient technologies are typified by a low-level of investment of time and effort in their productive process, and after use, the artefacts are quickly discarded [52]. In high-mobility contexts, curated artefacts are preferred in order to anticipate unexpected needs. In low-mobility contexts, the risk of food shortage is reduced and a strategy of expediency is favoured because artefacts are made on an on-needed basis [52].

Ethnographic data indicate that the composition of transported toolkits is dynamic and strongly influenced by the availability of resources and the tasks to be accomplished [52–59]. Oswalt [57,58] pointed out that toolkits for the acquisition of food are composed of different 'subsistants' that can be classed as instruments (e.g. sticks), weapons (e.g. harpoons, arrow points) and facilities (e.g. deadfall trap). The total amount of 'subsistants' indicates the diversity of the toolkit, and its complexity is measured by the total number of techno-units, that is, the different parts comprising a finished artefact [58]. Comparing the toolkits of several hunter–gatherer groups, Oswalt [58] argued that groups relying on hunting for their subsistence generally have more diversified and complex gear than those dependent on plant diets. However, placing hunting as the main activity of food acquisition can be a risky tactic because it requires more time for searching and chasing animal prey that may not ultimately be successfully captured. Torrence [59–61] hypothesized that as time–stress increases, hunter–gatherers tend to be more efficient, producing more specialized tools that enhance toolkit richness and complexity, and reduce the risk of failure. A similar hypothesis was advanced by Bleed [54], who explored several hunting toolkits and determined their diversity by applying the concepts of 'reliable' systems and 'maintenance' systems. While the maintenance approach generally has low failure costs because the gear can be easily turned into a different functional state, a reliable hunting kit must work when needed, and it is designed in such a way that function is guaranteed [54].

It has been noted, however, that this strict specialization of the toolkit limits the possibility of it being used for other tasks, suggesting that other tools need to be transported or crafted [54]. For example, the! Kung San [62], the Pumé [63], and Yanomano Amazonian hunters [64] craft light and portable weapons characterized by generalized tools that can be efficiently applied within a variety of circumstances without the need for *ad hoc* modifications. The hunting gear of the! Kung San and Yanomano are composed of poisoned arrows, spare arrowheads, and spears that do not change during the seasons or for different types of game [62,64]. A similar pattern is documented among the Pumé who often use spears as digging sticks, and arrowheads as butchering knives [63]. Conversely, the hunting gear of the Nunamiut is composed of more specialized tools produced in high number to fully exploit, during a short period, caribou migrations every year [65]. Shott [66] has pointed out that more generalized tools are used in groups that move frequently and in contrast with groups that move less frequently, as tools can be used for a broader range of tasks. Study of toolkit complexity among hunter–gatherers has indicated that technological richness tends to be correlated more with increased environmental risk and resource failure than with an increase in population size or residential mobility [67,68]. However, in certain conditions, these latter two variables may also have effects on cultural evolution [68,69].

Hunter-gatherers in tropical forests exhibit a great homogeneity in their technological responses for obtaining food [58,63,70–72]. Although poisoned arrows, blowpipes, and snares are used [63,71,73,74], hunting toolkits in tropical environments are thought to be less elaborate in comparison with forager gear in high latitudes [52,58,60]. This low elaboration pattern is present in the archaeological record; for example, though debated, the core and flake technology of the Hoabinhian of tropical Southeast Asia has been noted for its 'simplicity' [75–80]. The use of expedient technologies has often been explained as a result of either a loss of technical knowledge, the absence of fine-grained rocks, or the utilization of organic materials (e.g. wood, bamboo) which is unlikely preserved over the long-term [27,77,79,81,82]. According to the latter argument, the bamboo hypothesis was historically developed to explain the absence of handaxes in East Asia in the Middle Pleistocene [82,83]. Indeed bamboo residues have been found on some stone tools in archaeological contexts [84,85], and experimental work has demonstrated that bamboo may provide viable cutting edges, though stone artefacts are more efficient in keeping their edges sharper for a longer time [83,86].

Since the advent of processual archaeology, ethnographic data has been largely used as analogies for understanding the archaeological record (e.g. [65,87,88]). This approach fostered intense debates [89–92] and rejections as researchers [93–95] questioned to what extent the behaviours of contemporary hunter-gatherers, co-existing in modern economic systems, could reflect foraging and mobility patterns of past human communities [28,96]. Current views are that ethnographic analogies should be assessed on a context-by-context basis [89,97] or when cultural continuity between the archaeological and historic evidence can be demonstrated [98,99]. Recent investigations of Palaeolithic sites show that the composition of transported gear often diverges from ethnographic models [100–102]. Adaptive strategies to particular environments, including particular reactions to subsistence stress, were likely varied and not always operating optimality (e.g. [103–105]). From this perspective, exploitation strategies in rainforests during the Late Pleistocene and Holocene could have been wide-ranging. However, there is a gap in our knowledge about prehistoric settlement strategies in tropical forests. The well-preserved archaeological record in the rainforests of Sri Lanka provides an opportunity to examine the behavioural variability of *Homo sapiens* over time.

## Kitulgala Beli-lena rockshelter

Kitulgala Beli-lena is located *c*. 85 km east of Colombo in Sabaragamuwa Province, in the lowland Wet Zone of Sri Lanka (Fig 1). The rockshelter is a large (30 x 15 m) northwest-facing natural cavity formed from gneiss bedrock, part of the high-grade metamorphic terrain of Sri Lanka's Highland Complex [106]. Since the 1960s, several seasons of fieldwork has been carried out at the site revealing a long Late Pleistocene and Holocene archaeological sequence and an abundance of lithics [35,107,108]. In 2017, a new excavation was conducted through a collaboration between the Max Planck Institute for the Science of Human History (Jena, Germany) and the Department of Archaeology, Government of Sri Lanka (Fig 1). All relevant permits for this work were obtained from the Department of Archaeology, Government of Sri Lanka. The aim of the research was to apply multidisciplinary methods to refine the chronology of the sequence and to examine subsistence strategies and technical behaviours of foraging groups through time [34].

The excavation was situated in the inner western section of the rockshelter, about 5 m from the wall. Following the excavations in the rockshelter in 1985, the exposed sections were covered by stone walls to preserve the integrity of the site. The only portion that was not protected was a 2 m$^2$ excavation square (grid code: G12-G11) which was sampled for micromorphology

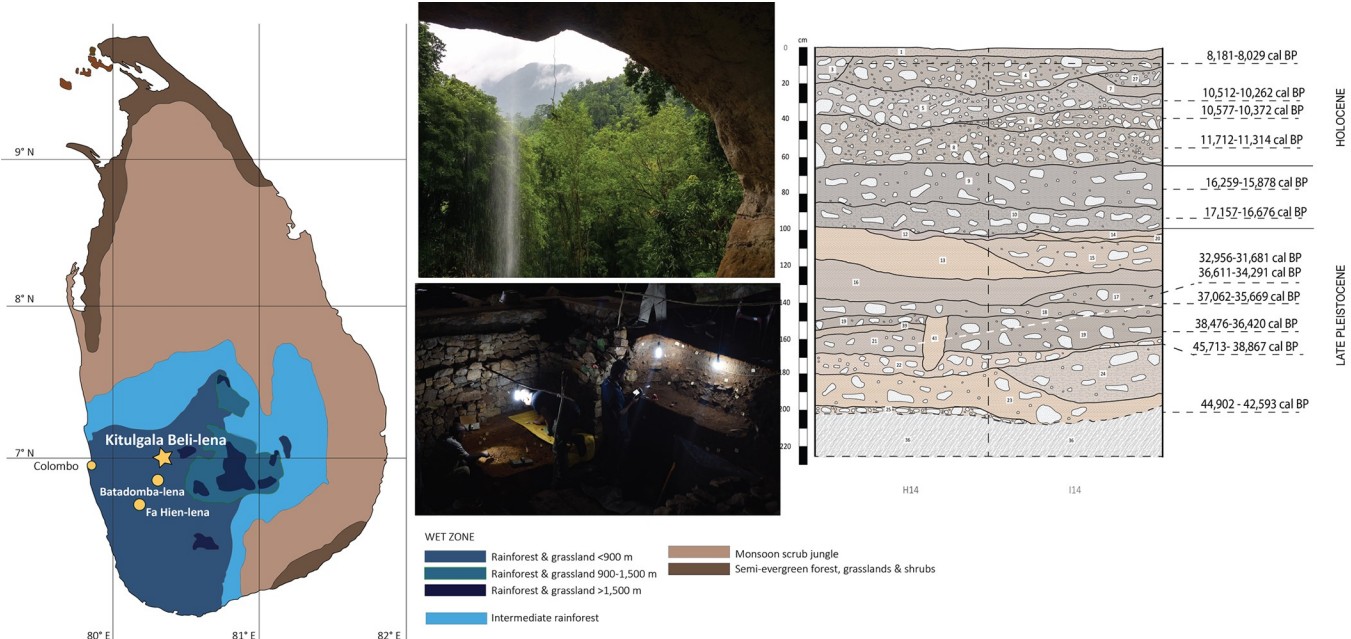

**Fig 1. Location of Kitulgala Beli-lena and other Sri Lankan rainforest sites mentioned in the text as well as the stratigraphic sequence from the 2017 excavation (base map modified from** https://commons.wikimedia.org/wiki/Category:SVG_topographic_maps_of_Sri_Lanka**; data of vegetation zones adapted from Ashton, P. S. & Gunatilleke, C. V. S. New light on the plant geography of Ceylon.** I. historical plant geography. J. Biogeogr. 14, 249–285 (1987), and Erdelen, W. Forest ecosystems and nature conservation in Sri Lanka. Biol. Conserv. 43, 115–135 (1988); photos taken by NA and stratigraphic section drawn by NA from original section drawings made by JB, AP and OW).

in 2005 and 2009 [44,108]. The new excavation extended this unprotected square southward, opening excavation square G12 and the previously unexcavated squares H14 and I14 (Fig 1).

The stratigraphic sequence of the site (Fig 1) has been divided into three main chronological phases [34]. The Late Pleistocene phase, dating between *c.* 45,000–31,000 years cal BP, is composed of 16 sub-horizontal layers of yellowish brown sandy clay to sandy silt sediments that overlay pebbly clayey loams with angular gneiss slabs atop the bedrock (context 36). At the bottom lies context 25, a firm mid yellowish brown (7.5 YR 6/6) clast supported conglomerate with well-rounded imbricated pebbles. This context may have been deposited by the stream that, at present, flows 60 m below the level of the cave entrance. Context 23 sits above, a firm yellowish brown (7.5 YR 6/8) sandy clay with presence of small to medium gneiss angular slabs. A radiocarbon date on a charcoal from this context yielded an estimate of 44,902–42,539 cal BP [34]. The sedimentary sequence continues with context 24, a compact mid brown (7.5 YR 5/4) sandy loam with the presence of medium to big angular gneiss slabs, overlain by context 22, a firm yellowish brown (7.5 YR 6/6) sandy loam with presence of medium to large gneiss slabs. This deposit was disturbed by termite bioturbation.

Context 21 is found on the left of square H14, a medium compact mid brown (7.5 YR 4/3) sandy clay with the presence of very few small gneiss slabs, overlain by context 39, a compact yellowish brown (7.5 YR 4/6) sandy clay, and context 19, a firm very dark brown (7.5 YR 2.5/3) sandy clay with the presence of small angular gneiss slabs that extend to the right side of square H14 and in square I14. In context 19, at the edge between H14 and I14 and on the NW quadrant of I14, two areas of fire use (respectively 0.20 m x 0.16 m; 15 m x 10 m) with some burning sediment, but without the presence of charcoal, ashes or burnt material were discerned. In this area, a small pit (cut 40) was recognized near the section truncating context 19, 39, 21 and 22. The pit has an extension of 0.23 m x 0.16 m, and it was filled with a yellowish

brown sandy clay (fill 41). In square G12, above context 19 lies context 26, a hard mid yellow-ish (7.5 YR 5/6) sandy clay with the presence of occasional gneiss slabs fallen from the cave roof.

In contrast, in squares H14 and I14, above context 19 the sequence continues with context 18, a mid compact dark brown (7.5 YR 3/4) sandy clay with the presence of small gneiss angular slabs, context 17, a firm dark yellowish brown (7.5 YR 3/4) with presence of gneiss slabs fallen from the cave roof, and context 16, a firm dark yellowish brown (7.5 YR 4/4) sandy clay. On top, is found context 13 comprised of a firm dark yellowish brown (7.5 YR 6/8) sandy clay, context 15, a firm dark yellowish brown (7.5 YR 5/6) sandy clay with the occasional presence of gneiss slabs, and context 20, a very firm dark yellowish brown (7.5 YR 5/8) sandy clay with the occasional presence of gneiss slabs fallen from the roof of the cave. The Late Pleistocene sequence ends with context 14, a firm pale greyish yellow (7.5 YR 5/6) sandy clay with little trace of anthropogenic activities, and context 12, a very firm mid orangish red (2.5 YR 4/4) silty clay. This context was a post-depositional altered clay similar to context 13 and 14, discolored and compacted by burning in overlying sediments.

The Terminal Pleistocene phase, dating between 17,157–11,314 years cal BP, comprises a stratigraphic succession of sub-horizontal layers of dark greyish brown sandy loam and silty clay. Context 10 is a moderate dark greyish brown (7.5 YR 2.5/2) sandy loam with the rare presence of gneiss spalls up to 10 cm in size, and occasional discrete clay mottles. Four potential hearths and some darker patches were also identified but without discrete charcoals. Context 9 is a moderate mid greyish brown (7.5 YR 3/4) sandy silt with the occasional presence of small/ medium size gneiss slabs fallen from the cave roof. The context was interpreted as a habitation deposit with abundant lithic material, almost all lying flat.

The Holocene Phase, dated 10,577–8,029 year cal BP, includes a compact mid-yellowish brown (7.5 YR 2.5/3) sandy loam layer with angular gneiss slabs at the bottom (context 8), superimposed by several nearly horizontal loamy clay and silty sand horizons [34,39,108]. Context 6, was a firm mid brown (7.5 YR 2.5/3) loamy clay with occasional inclusions of granite slabs, is overlain by context 5, a firm mid brownish yellow (7.5 YR 3/3) loamy clay including occasional granite slabs, and context 7, a firm mid greyish brown (7.5 YR 3/4) sandy silt, commonly sorted with granite chunks and grit. The sequence continues with context 27, a firm dark brown (7.5 YR 3/3) silty clay including 20% moderately sorted round pebbles and occasionally shells, followed by context 4, a loose mid brown (7.5 YR 5/3) silty sand with occasional snail shells and grit, and context 3, a loose mid brownish grey (7.5 YR 4/2) sandy silt supported by angular/ sub angular pebbles with occasional presence of shells and, rarely, with fine charcoals. The extraction of guano to be used as fertilizer on neighbouring rubber plantations during the colonial period has resulted in a 10 cm of silty sand deposit that caps the sequence (context 1) [39].

Archaeobotanical analysis from the site indicates a persistent use of wild breadfruit (*Artocarpus nobilis*) and possibly of kekuna nut (cf. *Canarium zeylanicum*) from the oldest layers [34]. Animal bone remains were documented only in the Holocene layers, and zooarchaeological and taphonomic studies confirm the exploitation of semi-arboreal and arboreal primates and squirrels at this time, a hunting behaviour common in the archaeological record of the other Pleistocene/Holocene sites on the island, such as at Fa-Hien Lena and Batadomba-lena [33,34,37]. Within the bone assemblage, several osseous tools (e.g. unipoints, bipoints and geometrics) were found and were probably used as projectile points, as seen at Fa-Hien Lena (Fig 2) [30,33,41]. A preliminary study of the lithic material showed some similarities with the assemblages of other Pleistocene sites in the Wet-Zone of Sri Lanka [34]. A deeper examination of the technological approaches used at Kitulgala Beli-lena allows a broader

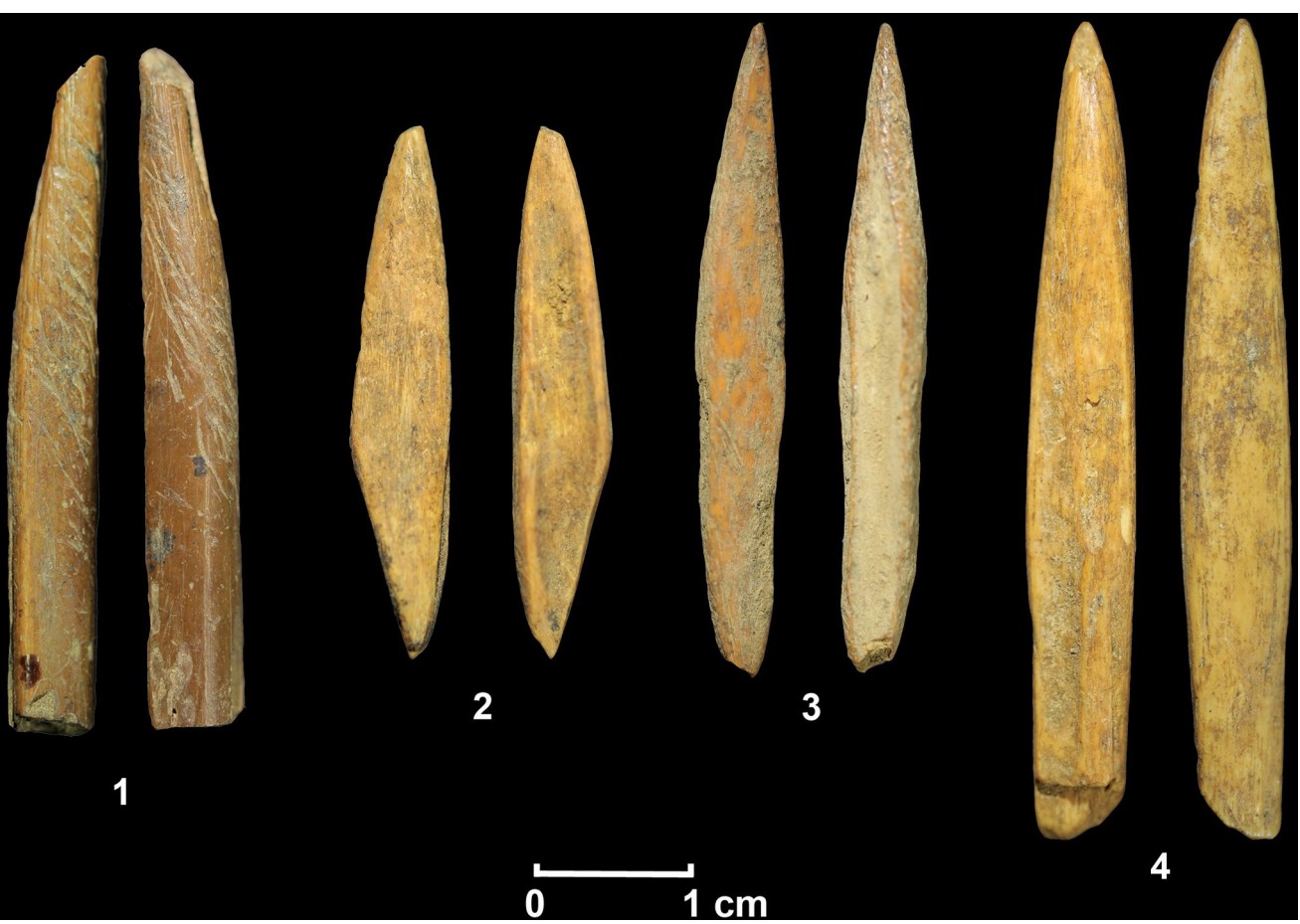

**Fig 2. Example of bone implements recovered from the Holocene layers of Kitulgala Beli-lena.** A- unipoint manufactured from cercopithecid fibula (context 3), B-D bipoints made from cercopithecid femur shaft fragments (B: context 5; C-D: context 27).

understanding of the processes of adaptability in the early and later phases of the site, and more widely, the exploitation of tropical environments in South Asia.

## Materials and methods

The aim of the current research is to examine technological trajectories present in Late Pleistocene and Holocene sites situated in the rainforests of Sri Lanka. Detailed technological studies have only been undertaken for stone tool assemblages recovered from Batadomba-lena [36,48] and Fa-Hien Lena [38] revealing lithic production oriented towards microlithic blanks. A preliminary report on the lithic assemblage of Kitulgala Beli-lena was published in Wedage and colleagues [34], though information on lithic production was not described. Here, we describe raw material selection strategies and knapping methods, using statistical analyses of metric attributes on cores and lithic by-products across different archaeological layers. The information drawn from Kitulgala allows for comparison with technical behaviours obtained from other archaeological sites in the rainforest zones of Sri Lanka, and more generally, Southeast Asia. Furthermore, the archaeological results may be compared to ethnographic data, with an aim to understand behavioural flexibility of *Homo sapiens* populations in rainforest habitats.

In this study, analysis is carried out on the lithic assemblages recovered from Kitulgala Beli-lena during the 2017 excavation season.

We analyse the lithic material following the *chaîne opératoire* concept, a methodological framework that defines the reconstruction of the various processes of flake production from the procurement of raw materials through the phases of manufacture and utilization until final discard [109,110]. The categories analysed include cores, flakes, bladelets, fragments and chips (fragments < 10mm). The presence of cortex on the lithic items was categorised into three different classes: cortical (>50% cortex), semi-cortical (>50% cortex), and no cortex.

We divide the quartz pebbles in five main categories based on their petrological features [128]: crystal–a translucent automorphic quartz; milky–a grainy xenomorphic quartz characterized by chalky/cloudy colour tonalities; rose–a grainy xenomorphic quartz characterized by light pink colour tonality; vein–a grainy xenomorphic quartz characterized by few reddish linear inclusions; granular–a course-grained xenomorphic quartz.

The metric attributes and the weights of all lithic artefacts were measured in order to assess the general features of the assemblage and to evaluate the degree of core reduction, *débitage* size, and discard thresholds. The maximum dimension and weight of the lithic items was measured and compared statistically using the free software PAST [111]. We first tested the normality of the data using Shapiro-Wilk tests (S3 Table); based on these results we employed a non-parametric test (Mann Whitney test, $\alpha = 0.05$) to estimate the difference in average weight and maximal size between cores and blanks among the different raw materials. When the number of artefacts was lower than 20, descriptive statistics were used for comparisons. We also compared the relations between the length and the weight of different core categories by chronological phase using a linear regression model.

Here, we use the term 'microlithic' to refer to lithic knapping methods aimed at the production of small blanks, following the criteria proposed by Pargeter and Shea [112], rather than limiting the definition to small and backed artefacts (geometric or non-geometric). Following the demonstration of the bimodal distribution of blade sizes in southern India [113], we employ a 40 mm size threshold and describe flakes, blades (bladelets), and retouched tools smaller than 40 mm as microlithic (see [38]).

Previous technological studies of Pleistocene lithic assemblages from Sri Lanka document the use of the bipolar-on-anvil (henceforth bipolar) and freehand methods [36,38]. In this study, bipolar knapping is interpreted as the technique of placing a core with a bare hand on a stationary anvil and striking it with a hammerstone in perpendicular planes from the top. The force applied from the hammerstone produces two opposed impact points, one on the upper face of the core and the second on the lower face that is in contact with the anvil. Since in this percussion technique (bipolar *sensu stricto*) the core is perpendicular to the anvil, flakes are produced by the hitting of the hammerstone with the upper face and by the counterstrikes of the core with the anvil [114].

The morphology of the nodules among different raw materials can influence the position of how a pebble is placed on an anvil (Fig 3). In this study, two modalities are distinguished: a) vertical axial knapping, when the pebble is oriented along the longer axis, and b) horizontal axial knapping, when the pebble is oriented along his shorter axis. During reduction events, the striking angle tends to be ~90° although some variations in the angulation between the hammer and the striking platform can be produced due to fractures, rotation of the core, or re-organization of the core volume [115]. In this case, the vertical and horizontal modalities are classified as non-axial.

The striking of the hammerstone with the proximal surface of the core could produce battering marks, a hertzian cone, a linear striking platform complemented by scaled or invasive bifacial detachments, or a pointed striking platform [114–117]. The hitting of the core with a hard anvil produces splintering and pointed platforms, as well as battering marks. If the anvil used is soft (e.g. wood) or the core is resting simply on the ground, opposing bulb or other

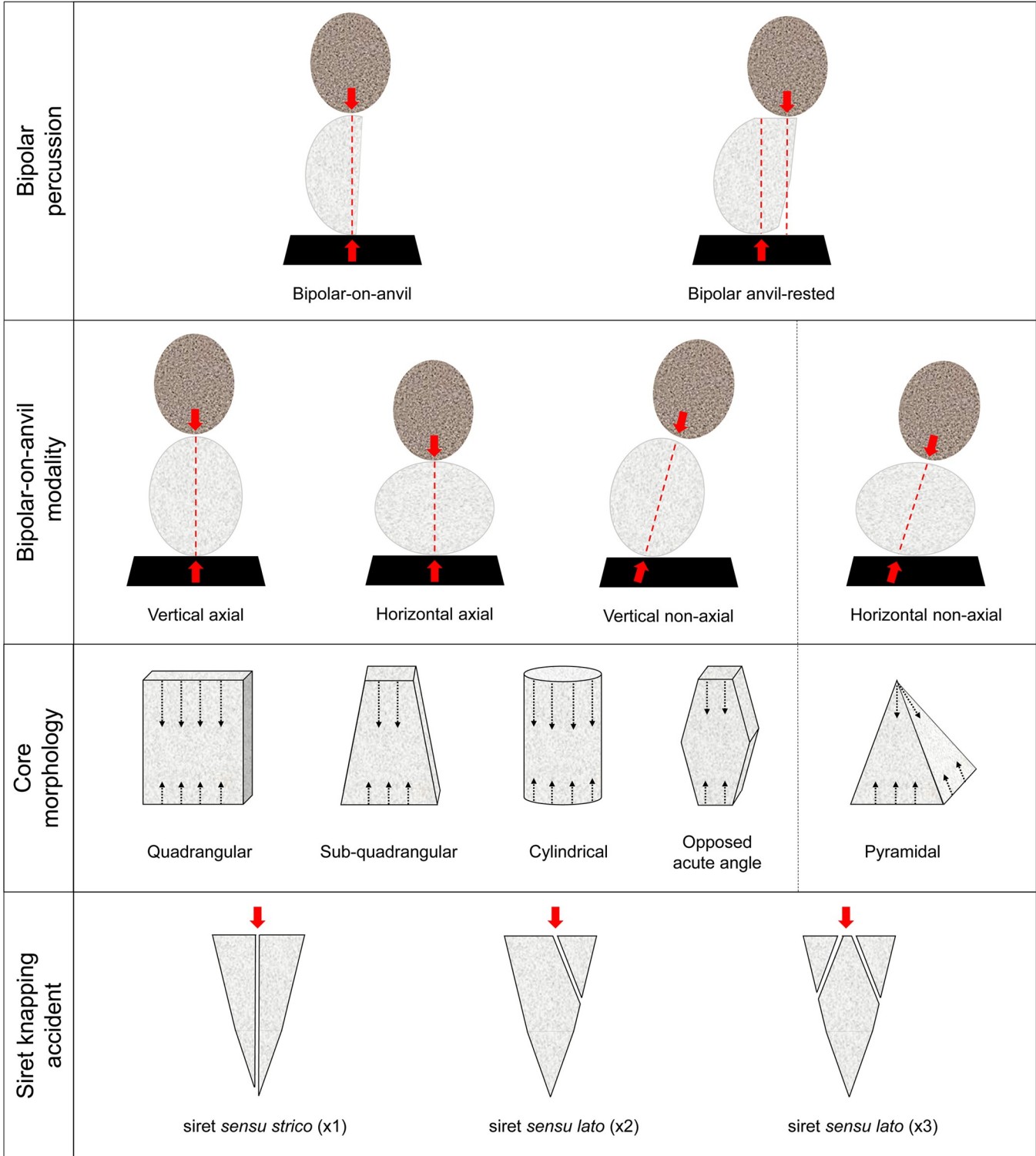

**Fig 3. Schematic model of the reduction systems and the terminology used in this paper.**

counterblow fractures will be absent [118,119]. Beyond the recurrent quadrangular, sub-quadrangular and cylindrical forms with opposing scars, the core morphologies could vary on the basis of the degree of reduction resulting in: a) a core without a proper striking platform and characterized by two opposed acute angles, or b) a core with a pointed percussion point and flat lower surface, generally resulting from a horizontal no-axial reduction [117,120,121].

The anvil-rested modality is distinguished when the core is only stabilized on the anvil before the flake detachment, but it is missing the higher compressive stress and degree of immobilization, characteristic of bipolar reduction [122–124]. In this manner, the point of contact between the hammer/core and core/anvil may not aligned.

The bipolar technique differs from freehand knapping in terms of fracture mechanics, and in the former this includes wedging initiations, compression–propagation and preferential axial terminations [125]. In terms of direction of the detachments, freehand reduction is often secant to one of the core axes and, when it is parallel, it requires the preparation of a striking platform [126,127]. The identification of by-products of bipolar percussion has been a matter of debate [116,118,127–133]. The main issue is that bipolar knapping does not focus on controlling blank production, thereby producing a large volume of lithic by-products (e.g. basal, parasitic or irregular flakes and fragments). Generally, bipolar flakes show various different morphologies and sizes in comparison to freehand percussion, characterized by diffuse bulbs of percussion, shattered platforms and opposed fracture edges (e.g. hinge, step). Typical by-products of bipolar reduction include *bâtonnet* flakes (or bipolar spalls), non-cortical flakes with longitudinal fractures and triangular/quadrangular sections [134], splinter flakes, [128] and siret knapping accidents. Splinter flakes are not considered retouched artefacts but pieces retaining small portions of the core on distal sides that are characterized by more or less pronounced traces of longitudinal fracture [128]. Siret knapping accidents were distinguished following the criteria of Mourre [115] (Fig 3): a) siret *sensu strico* (x1) is considered a fracture parallel to the flaking axis that divides the blank in two parts, more or less equal; b) siret *sensu lato* (x2) is considered a fracture that removes a portion of the flake proximal side secant to the direction of the flaking axis; c) siret *sensu lato* (x3) is considered a fracture that removes two opposed portions of the flakes proximal side obliquely to the direction of the flaking axis. Often, the remaining part of the platform shows a pointed morphology.

Analysis of lithic products and by-products from Kitulgala Beli-lena is examined by chronological phase, i.e., sub-divided into Late Pleistocene, Terminal Pleistocene, and Holocene categories.

## Results

### Late Pleistocene (c. 45,000–31,000 cal BP)

The Late Pleistocene lithic assemblage includes 5,012 artefacts (4,992 lithic items and 20 hammers) (Tables 1, S1 and S2). The raw material used most is crystal quartz, followed by milky quartz and other quartz varieties in lower percentages (Table 1, Fig 4). Two flakes and four fragments of chert are also found in the collection (Table 1). Technological analysis of cores indicates that the main knapping strategy used is the bipolar method (Table 2, Fig 5). Generally, heterogeneous quartz pebbles of different sizes were collected for flake production. In only two cases (milky quartz, crystal quartz) were the nodules probably tabular blocks. In these latter cases, the striking platforms and the distal surfaces are flat, and production was carried out on one lateral flaking surface. Although the shape of the blocks allows a secure contact with the anvil during vertical knapping procedures, these cores were not fully exploited, and were abandoned, although flake production could have continued. In the remaining assemblage, two cores in crystal quartz, five cores in milky quartz and one in rose quartz are partially

**Table 1. Total number and percentages of lithic artefacts in the Late Pleistocene.**

| | Crystal | % | Milky | % | Veiny | % | Rose | % | Grainy | % | Chert | % | Total | % |
|---|---|---|---|---|---|---|---|---|---|---|---|---|---|---|
| **Cortical flake >50%** | 8 | 0.3 | 18 | 1 | | | 4 | 2 | 2 | 0.3 | | | 32 | 0.6 |
| **Cortical flake <50%** | 50 | 2.1 | 32 | 1.8 | 1 | 12.5 | 14 | 6.9 | 8 | 1.3 | | | 105 | 2.1 |
| **Cortical core-edge flake** | 1 | 0.05 | 1 | 0.1 | | | 1 | 0.5 | | | | | 3 | 0.1 |
| **Flake** | 254 | 10.5 | 106 | 6.1 | | | 4 | 2 | 21 | 3.5 | 2 | 33.3 | 387 | 7.8 |
| **Splinter flake** | 7 | 0.3 | 6 | 0.3 | | | | | | | | | 13 | 0.3 |
| **Core-edge flake** | | | 1 | 0.1 | | | | | | | | | 1 | 0.02 |
| **Bladelet** | 2 | 0.1 | | | | | | | | | | | 2 | 0.04 |
| **Cortical fragment** | 52 | 2.1 | 83 | 4.8 | | | 17 | 8.3 | 18 | 3 | 1 | 16.7 | 171 | 3.4 |
| Siret x1 | 5 | 0.2 | 9 | 0.5 | | | 1 | 0.5 | 1 | 0.2 | | | 16 | 0.3 |
| Siret x2 | 3 | 0.1 | 2 | 0.1 | | | | | | | | | 5 | 0.1 |
| Siret x3 | | | 1 | 0.1 | | | | | 2 | 0.3 | | | 3 | 0.1 |
| **Fragment** | 878 | 36.2 | 617 | 35.4 | 4 | 50 | 94 | 46.1 | 316 | 52 | 3 | 50.0 | 1912 | 38.3 |
| Siret x1 | 101 | 4.2 | 41 | 2.4 | 1 | 12.5 | 3 | 1.5 | 17 | 2.8 | | | 163 | 3.3 |
| Siret x2 | 11 | 0.5 | 7 | 0.4 | | | | | | | | | 18 | 0.4 |
| Siret x3 | 20 | 0.8 | 11 | 0.6 | | | 2 | 1 | 4 | 0.7 | | | 37 | 0.7 |
| **Chips** | 983 | 40.6 | 742 | 42.6 | | | 59 | 28.9 | 214 | 35.2 | | | 1998 | 40 |
| **Core** | 20 | 0.8 | 35 | 2 | 1 | 12.5 | 2 | 1 | 1 | 0.2 | | | 59 | 1.2 |
| **Core fragment** | 29 | 1.2 | 30 | 1.7 | 1 | 12.5 | 3 | 1.5 | 4 | 0.7 | | | 67 | 1.3 |
| **Total** | 2424 | 100 | 1742 | 100 | 8 | 100 | 204 | 100 | 608 | 100 | 6 | 100 | 4992 | 100 |

fractured on the distal and/or proximal sides, probably due to internal flaws or to the impact of heavy hammerstones. Although the knapping modality applied could be identified, these artefacts are incomplete and were excluded in the subsequent statistical comparison.

The majority of cores are reduced along their longest axis followed by the vertical axial modality (Table 2); 43% of the artefacts retain a portion of cortex on their dorsal side. Compression between the hammerstones and the anvils created cores characterized by bidirectional detachments on the flaking surfaces (Fig 5). Among seven cores (crystal n = 2, milky n = 4, grainy n = 1), flake production is limited to the ventral surface. In other samples, the battering of the hammerstone and recurrent counterstrikes with the anvil also produced reduction

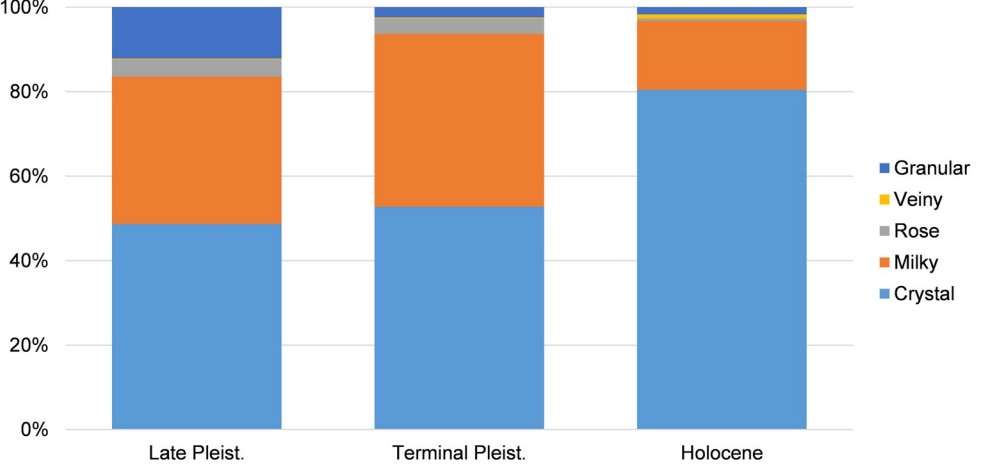

**Fig 4. Frequency of quartz types by chronological phases recorded at Kitulgala Beli-lena.**

**Table 2. Total number and percentages of the types of cores by different raw materials in the Late Pleistocene phase.**

| | Crystal | % | Milky | % | Veiny | % | Rose | % | Grainy | % | Total | % |
|---|---|---|---|---|---|---|---|---|---|---|---|---|
| **Vertical** | 11 | *55* | 24 | *68.6* | 1 | 100 | 2 | 100 | 1 | 100 | 39 | *66.1* |
| **Horizontal** | 4 | *20* | 3 | *8.6* | | | | | | | 7 | *11.9* |
| **Horizontal non-axial** | 1 | *5* | 2 | *5.7* | | | | | | | 3 | *5.1* |
| **Orthogonal** | 3 | *15* | 5 | *14.3* | | | | | | | 8 | *13.6* |
| **Anvil-rested** | | | 1 | *2.9* | | | | | | | 1 | *1.7* |
| **Unidirectional** | 1 | *5* | | | | | | | | | 1 | *1.7* |
| **Total** | 20 | *100* | 35 | *100* | 1 | 100 | 2 | 100 | 1 | 100 | 59 | *100* |

schemes on the dorsal surface. Among two crystal quartz and milky quartz cores, the distal surface was flat, created probably by previous detachments or fractures. The cores were most likely rotated on the flat side in order to take advantage of a more stable surface on the anvil. However, evidence of previous reduction sequences are absent, suggesting that the cores were reduced intensively after rotation and the use of new striking platforms. Although crushing, fractures and hertzian cones are common in the assemblage, it is worth noting a core in crystal quartz characterized by a denticulate delineation of the distal side. This feature is often related to uneven resting or eccentric impact points that caused a failure in detaching the counter-strike flakes.

With the exception of two large artefacts in milky quartz (layer 34, n°: 319—weight: 183.8 g; length: 55 mm, width: 63.8 mm, thickness: 65.5 mm; layer 35, n° 1259—weight: 423.6 g; length: 71.1 mm, width: 92.3 mm, thickness: 39.4 mm), comparison of the mean values between crystal and milky vertical axial cores reveals no obvious differences (Table 3). A similar pattern is found comparing the mass and size of a grainy core (weight: 15.3 g; length: 30.2 mm, width: 24.6 mm, thickness: 17.1 mm) whereas the veiny (weight: 3.4 g; length: 19.7 mm, width: 17.5 mm, thickness: 11.8 mm) and rose cores (mean values: weight: 0.9 g; length: 18.6 mm, width: 8.2 mm, thickness: 7 mm) are smaller.

The second largest technological group in the assemblage includes cores reduced along their smaller axis (horizontal) (Table 2). In seven cores (crystal n = 4, milky n = 3), the resting point is located on the vertical line in relation to the percussion point (axial). In this group, four artefacts (crystal n = 3, milky n = 1) are characterized by a flat distal surface indicating that this particular feature of the cores was pursued by the knappers regardless of the size of the by-products. Previous reduction events were absent in the flaking surfaces. In the remaining three cores of the assemblage (crystal n = 1, milky n = 2) (Table 2), the resting and contact points are off-axis producing an uneven exploitation of the nodule. The distal surfaces are flat whereas the proximal sides are pointed yielding a pyramidal morphology. Descriptive statistics suggest no substantive difference between the weight and size of the cores in crystal and milky quartz (Table 3).

In a few examples, bipolar cores were rotated 90° degrees, producing orthogonal scars on the flaking surface and shaping the outline into a quadrangular morphology (Table 2). In four cores (crystal n = 1, milky n = 3), the rotation was promoted by a fracture along the knapping axis (siret-type accident) and this latter axis was then used as new distal surface. In the other cores (crystal n = 2, milky n = 2), orthogonal reduction was more recurrent with several changes of the striking platform. Comparison of the metric attributes indicate no difference between the artefacts in crystal and milky quartz (Table 3). Although the orthogonal modality is thought to reduce the core volume greatly due to the exploitation of different striking platforms, the length and width values of orthogonal cores falls back within the range of variability of vertical and horizontal artefacts (Fig 6). Moreover, the linear regression model reveals that,

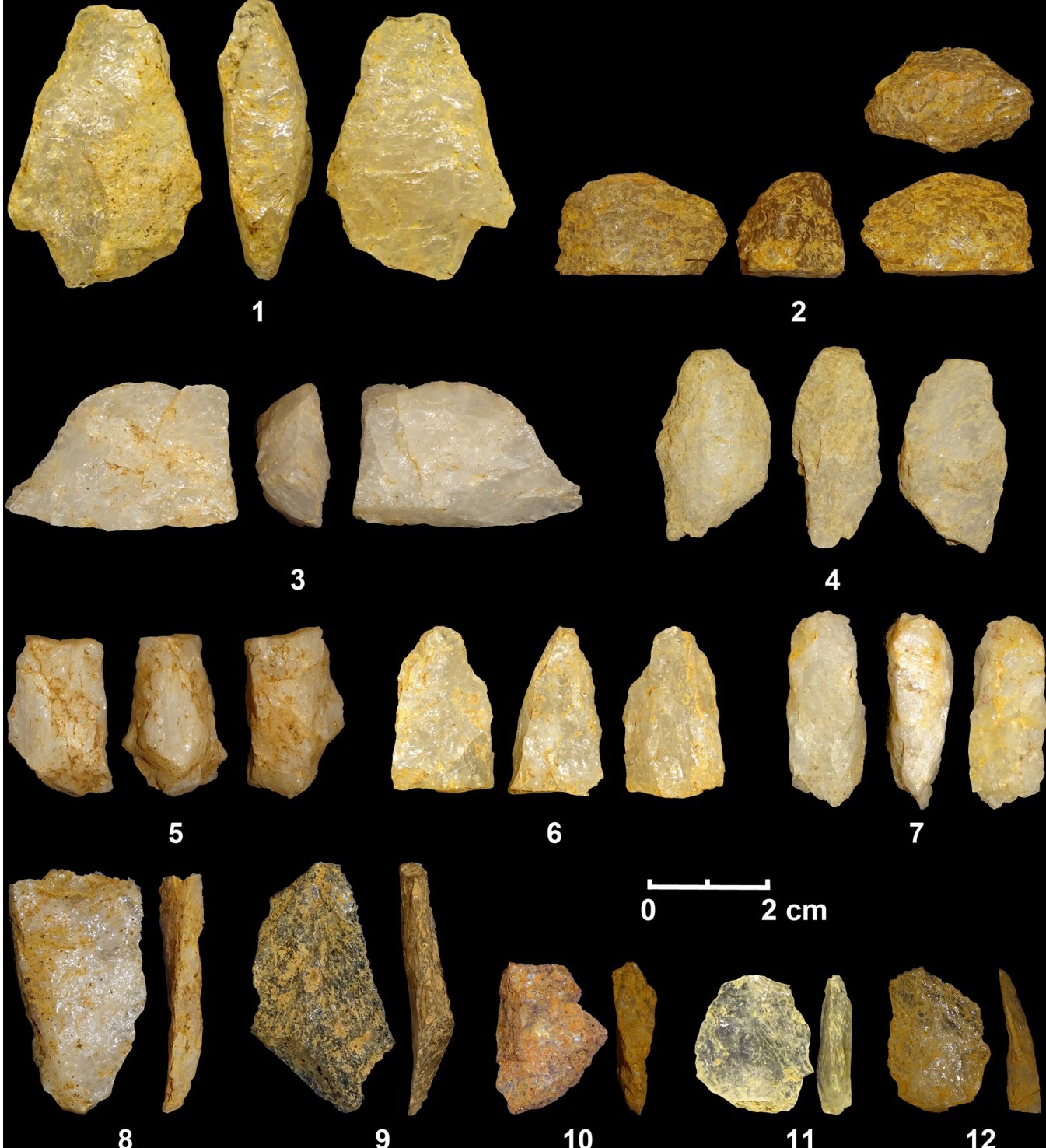

**Fig 5. Bipolar cores (1–7) and flakes (8–12) from the Late Pleistocene contexts of Kitulgala Beli-lena.**

in similar length values, the cores reduced with the vertical modality are weightier than the others although the difference between the slopes is not significant (F = 0.707521, DFn = 2, Dfd = 41, *p* = 0.4988) (Fig 7).

**Table 3. Counts, mean (μ) and standard deviation (σ) of the metric attribute (mm) of the bipolar-on-anvil cores of Kitulgala Beli-lena in the Late Pleistocene phase.**

| | Attributes | Crystal | | | Milky | | |
|---|---|---|---|---|---|---|---|
| | | N° | μ | σ | N° | μ | σ |
| **Vertical** | Weight | 10 | 18.9 | *22* | 17 | 19.2 | *17.8* |
| | Length | | 26.7 | *6.3* | | 29.5 | *4.8* |
| | Width | | 25.7 | *9.4* | | 26.1 | *7.2* |
| | Thickness | | 17.4 | *7.5* | | 19.5 | *6.3* |
| **Horizontal** | Weight | 4 | 17.3 | *10.1* | 5 | 40.7 | *25.1* |
| | Length | | 28.4 | *1.5* | | 37.7 | *8.9* |
| | Width | | 25.8 | *8.4* | | 32.6 | *8* |
| | Thickness | | 18.9 | *3.8* | | 24.6 | *7.6* |
| **Orthogonal** | Weight | 3 | 17 | *4.6* | 5 | 30.1 | *34.4* |
| | Length | | 31.5 | *8.2* | | 31.6 | *15.3* |
| | Width | | 28.1 | *7.7* | | 33.4 | *17.1* |
| | Thickness | | 18.6 | *3* | | 26.7 | *18.1* |

An anvil-rested core in milky quartz and a unidirectional core were present in the assemblage (Table 2, S2 Fig). The former artefact (weight: 116.8 g; length: 49.3 mm, width: 55.2 mm, thickness: 33.6 mm) is characterized by an initial knapping event that split the pebble in two pieces, and then the ventral surface was used as new striking platform for the production of three flakes. The analysis of the scars and striking platform support the hypothesis that the core was reduced using the rested on anvil modality. The freehand core is, in contrast, a large pebble (weight: 388 g; length: 91.1 mm, width: 55.6 mm, thickness: 45.1 mm) that was firstly decorticated, and then discarded after the detachment of one flake.

Analysis of the cortical flake assemblage indicates that the phases of decortication were carried out on site for most of the quartz varieties (Table 1). Semi-cortical flakes are more numerous whereas the cortical core-edge flakes are limited (Table 1). Splinter pieces are recorded in one cortical flake in rose quartz, three semi-cortical flakes in clear quartz, and in three cortical fragments. The data on crystal and milky quartz show the largest frequencies of cortical and unbroken flakes (Table 1). In this latter category, the diacritic reading of the scars on the dorsal surfaces reveals that most of the blanks are characterized by a bidirectional (53.4%) and unidirectional pattern (46%) whereas the orthogonal examples (0.6%) are found only in crystal and milky quartz flakes. The technological features of the flakes suggest that nearly the totality of the blanks were by-products of the bipolar method (Fig 5) whereas only two flakes could be associated with freehand reduction. Together with flakes, other typical by-products of the bipolar technology are also found such as splinter flakes and siret knapping accidents

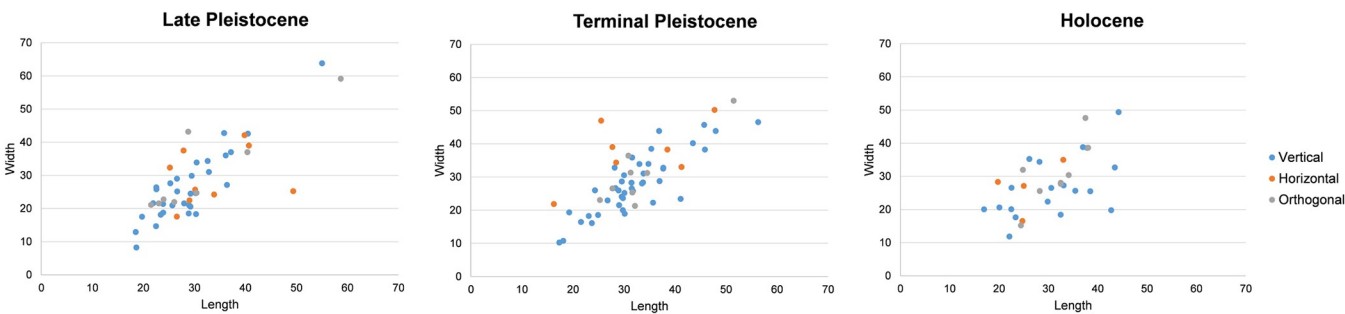

**Fig 6. Plot of the relation between the length (mm) and the width (mm) of different categories of cores of Kitulgala Beli-lena by chronological phases.**

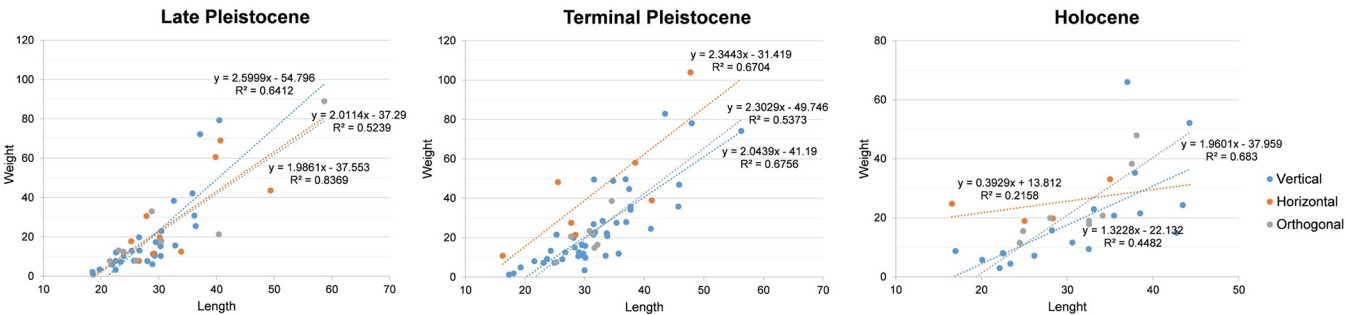

**Fig 7. Plot of the relation between the length (mm) and the weight (gr) of different categories of cores of Kitulgala Beli-lena by chronological phases.**

(Table 1). The fractures siret *sensu stricto* (x1) are common in the different quartz varieties whereas the other siret breakages *sensu lato* are more frequent in crystal and milky quartz (Table 1).

Comparison of the length distribution shows that the bulk of the flake assemblage is smaller than 20 mm (Fig 8, Tables 4 and 5). A statistical comparison reveals a significant difference in the median values of cortical flakes in crystal and milky quartz (Mann–Whitney test, $p = 0.0026$), and in crystal and rose quartz (Mann–Whitney test, $p = 0.0289$), and between complete flakes in crystal and milky quartz (Mann–Whitney test, $p = 0.0202$).

## Terminal Pleistocene (17,157–11,314 cal BP)

The lithic assemblage of the Terminal Pleistocene phase is comprised of 6,577 artefacts (6,571 lithic items and 6 hammers) (Tables 6, S1 and S2). The primary raw materials used are crystal quartz and milky quartz (Fig 4). Four complete flakes, one fragment, two chips and one core-

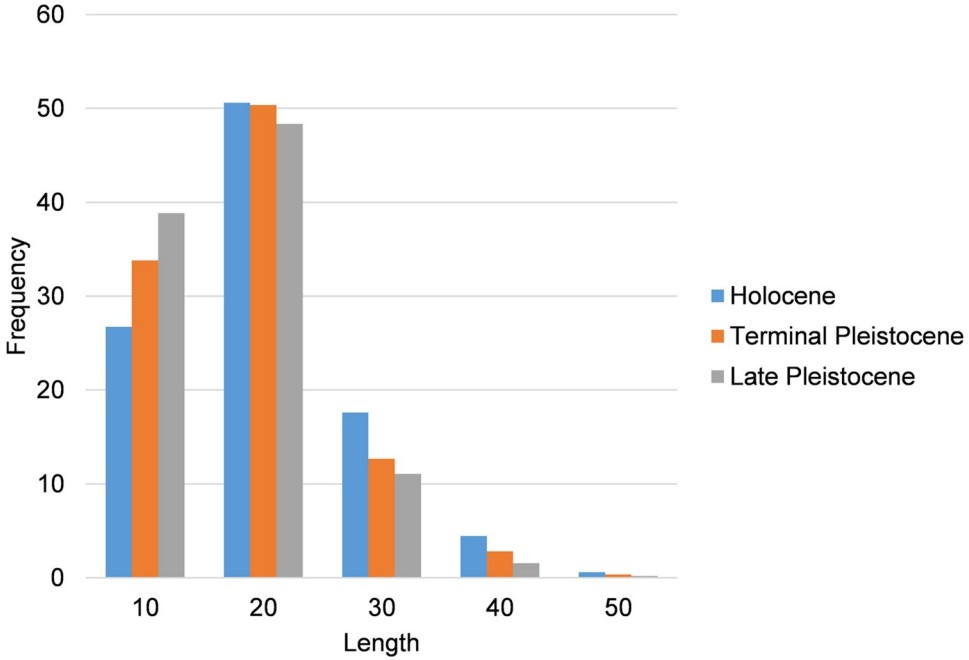

**Fig 8. Histogram of the frequency of complete flakes by length intervals during the different chronological phases at Kitulgala Beli-lena.**

**Table 4. Count, mean (μ) and standard deviation (σ) of the length (mm) of cortical flakes in Kitulgala Beli-lena.**

| Phase | Crystal | | | Milky | | | Veiny | | | Rose | | | Grainy | | |
|---|---|---|---|---|---|---|---|---|---|---|---|---|---|---|---|
| | N° | μ | σ | N° | μ | σ | N° | μ | σ | N° | μ | σ | N° | μ | σ |
| Holocene | 45 | 22.6 | 7.2 | 43 | 27.9 | 12.2 | 2 | 28.1 | 6.3 | 1 | 26.5 | | 2 | 35.86 | 18.1 |
| Terminal Pleistocene | 45 | 21.2 | 6.9 | 92 | 25.5 | 9.3 | 1 | 27.3 | | 7 | 16.7 | 3.3 | 1 | 31.4 | |
| Late Pleistocene | 59 | 18.4 | 5.6 | 51 | 22.6 | 7.3 | 1 | 21.3 | | 19 | 22.9 | 9.4 | 10 | 19.9 | 7.3 |

on-flake in chert were also present in the collection (Table 6). Three cores in clear quartz and four cores in milky quartz are partially fractured on the proximal and/or distal sides and are excluded from the descriptive statistical analysis. Technological analysis indicates that the main flaking method used was the bipolar in the vertical axial modality (Table 7, Fig 9). Two cores in milky quartz show production limited to the flaking surface, whereas the dorsal sides are cortical. Conversely, the other cores present a reduction of the dorsal sides owing battering of the hammerstone and counterstrikes on the anvil; 36% of the cores (crystal n = 3, milky n = 13, rose n = 1) retain small portions of cortex. Within common examples of the modality vertical axial, four cores in crystal quartz and four in milky quartz are characterized by a flat distal side, probably caused by previous detachments or by vertical fractures similar to siret-type accidents. The opportunistic use of the flat surface was probably beneficial for a more stable contact of the core on the anvil. In another two cores in crystal and milky quartz, the striking platforms are battered and, although the use of other striking platforms were not detected, the final morphologies are sub-quadrangular.

The core-on-flake in chert (weight = 10.8 g, length = 33.5 mm, width = 28 mm, thickness = 9.59 mm) is a cortical blank characterized by the three unidirectional removals in the ventral surface, probably detached using the bipolar vertical axial modality.

The other knapping modality applied is horizontal (Table 7). In six artefacts (crystal n = 2, milky n = 4), the reduction pattern is axial, with four cores (crystal n = 1, milky n = 3) characterized by a flat distal surface in contact with the anvil. In this group, 40% of the blanks present portions of cortex. In three other artefacts (crystal n = 2, milky n = 1) (Table 7), the directions of the point of contact hammerstone/core and core/anvil were not aligned, favouring the development of artefacts with sub-pyramidal morphologies and elongated removals.

Another reduction modality applied to seven artefacts (crystal n = 1, milky n = 6) is orthogonal axial, characterized by recurrent bipolar flake production and the rotation of the flaking surfaces 90˚ degrees for better exploitation of the core volume. Changes in the striking platform shaped the cores into quadrangular morphologies, and some of them preserve traces of cortex (crystal n = 1, milky n = 2). One artefact in milky quartz is very large (weight = 173.8 g, length = 51.5 mm, width = 53 mm, thickness = 44.2 mm) in comparison to the other samples, which is excluded in the descriptive statistical analysis as it is an outlier. One artefact in crystal and one in milky quartz in the assemblage shows the presence of a flat distal side, produced by a previous knapping accident similar to the siret-type breakage. As in previous examples from the vertical and horizontal axial groups, the use of the flat distal side was opportunistic and

**Table 5. Count, mean (μ) and standard deviation (σ) of the length (mm) of complete flakes in Kitulgala Beli-lena.**

| Phase | Crystal | | | Milky | | | Veiny | | | Rose | | | Grainy | | | Chert | | |
|---|---|---|---|---|---|---|---|---|---|---|---|---|---|---|---|---|---|---|
| | N° | μ | σ | N° | μ | σ | N° | μ | σ | N° | μ | σ | N° | μ | σ | N° | μ | σ |
| Holocene | 301 | 20 | 6.7 | 105 | 24.2 | 8.1 | | | | 2 | 34.1 | 16.8 | 2 | 21.1 | 6.1 | 3 | 26.1 | 6.2 |
| Terminal Pleistocene | 208 | 18.3 | 5.2 | 206 | 21.4 | 7.2 | 1 | 15.9 | | 2 | 15.5 | 4.7 | 1 | 23.5 | | 3 | 22.7 | |
| Late Pleistocene | 245 | 19 | 6.8 | 106 | 20.5 | 6.7 | | | | 4 | 18.7 | 2.6 | 21 | 22.7 | 9.7 | 2 | 19.4 | 5.2 |

**Table 6. Total number and percentages of lithic artefacts in the Terminal Pleistocene.**

| | Crystal | % | Milky | % | Veiny | % | Rose | % | Grainy | % | Chert | % | Total | % |
|---|---|---|---|---|---|---|---|---|---|---|---|---|---|---|
| **Cortical flake >50%** | 10 | 0.3 | 28 | 1 | 1 | 5.9 | 2 | 0.8 | 1 | 0.6 | | | 42 | 0.6 |
| **Cortical flake <50%** | 35 | 1 | 64 | 2.4 | | | 5 | 2.1 | | | 1 | 12.5 | 105 | 1.6 |
| **Flake** | 213 | 6.2 | 210 | 7.8 | 1 | 5.9 | 2 | 0.8 | 2 | 1.3 | 3 | 37.5 | 431 | 6.6 |
| **Splinter flake** | 3 | 0.1 | 10 | 0.4 | | | | | | | | | 13 | 0.2 |
| **Core-edge flake** | | | 1 | 0.04 | | | | | | | | | 1 | 0.02 |
| **Bladelet** | 1 | 0.03 | | | | | | | | | | | 1 | 0.02 |
| **Bâtonnet** | 1 | 0.03 | | | | | | | | | | | 1 | 0.02 |
| **Cortical fragment** | 61 | 1.8 | 107 | 4 | 4 | 23.5 | 7 | 2.9 | 2 | 1.3 | | | 181 | 2.8 |
| **Siret x1** | 4 | 0.1 | 9 | 0.3 | | | | | | | | | 13 | 0.2 |
| **Siret x2** | 2 | 0.1 | 2 | 0.1 | 1 | 5.9 | | | | | | | 5 | 0.1 |
| **Siret x3** | | | 2 | 0.1 | | | | | | | | | 2 | 0.03 |
| **Fragment** | 1038 | 30 | 1045 | 38.9 | 8 | 47.1 | 73 | 30.7 | 65 | 40.6 | 1 | 12.5 | 2230 | 33.9 |
| **Siret x1** | 87 | 2.5 | 87 | 3.2 | 1 | 5.9 | | | 3 | 1.9 | | | 178 | 2.7 |
| **Siret x2** | 6 | 0.2 | 5 | 0.2 | | | | | | | | | 11 | 0.2 |
| **Siret x3** | 6 | 0.2 | 8 | 0.3 | | | | | | | | | 14 | 0.2 |
| **Chips** | 1971 | 56.9 | 1028 | 38.3 | | | 143 | 60.1 | 86 | 53.8 | 2 | 25 | 3230 | 49.2 |
| **Core** | 20 | 0.6 | 46 | 1.7 | | | 1 | 0.4 | | | 1 | 12.5 | 68 | 1 |
| **Core fragment** | 3 | 0.1 | 35 | 1.3 | 1 | 5.9 | 5 | 2.1 | 1 | 0.6 | | | 45 | 0.7 |
| **Total** | 3461 | 100 | 2687 | 100 | 17 | 100 | 238 | 100 | 160 | 100 | 8 | 100 | 6571 | 100 |

limited to one striking platform; in this assemblage, the presence of even distal surfaces in two cores are combined with orthogonal detachments.

Comparison of metric attributes indicates no significant difference between the mean values between crystal and milky cores knapped using vertical and horizontal modalities (Table 8). However, the distribution of the length and width values demonstrates that vertical axial artefacts have the broadest variability (Fig 6) and a significant correlation exists between the length and width values (n = 40, r = 0.7816, p = $\leq$ 0.0001). Conversely, orthogonal bipolar cores are clustered in the range between 20–40 mm whereas horizontal artefacts are scattered and less standardized (Fig 6). The linear regression model shows that with similar length values, horizontal cores are weightier than the others, even if the difference between the slopes is not significant (F = 0.283224, DFn = 2, Dfd = 52, *p* = 0.7545) (Fig 7).

Analysis identified two exhausted bipolar vertical axial cores (crystal n = 1, milky n = 1), and before discard, they were exploited opportunistically for short flake production sequences. In the first crystal quartz example (weight: 12.5 g; length: 26.7 mm, width: 22.9 mm, thickness: 16.9 mm), the striking platform fractured during reduction and a final unidirectional flake was

**Table 7. Total number and percentages of the types of cores by different raw materials in the Terminal Pleistocene phase.**

| | Crystal | % | Milky | % | Rose | % | Chert | % | Total | % |
|---|---|---|---|---|---|---|---|---|---|---|
| **Vertical** | 14 | 70 | 32 | 69.6 | 1 | 100 | 1 | 100 | 48 | 70.6 |
| **Horizontal** | 2 | 10 | 4 | 8.7 | | | | | 6 | 8.8 |
| **Horizontal non-axial** | 2 | 10 | 1 | 2.2 | | | | | 3 | 4.4 |
| **Orthogonal** | 1 | 5 | 6 | 13 | | | | | 7 | 10.3 |
| **Bipolar + unid.** | 1 | 5 | 1 | 2.2 | | | | | 2 | 2.9 |
| **Unidirectional** | | | 2 | 4.3 | | | | | 2 | 2.9 |
| **Total** | 20 | 100 | 46 | 100 | 1 | 100 | 1 | 100 | 68 | 100 |

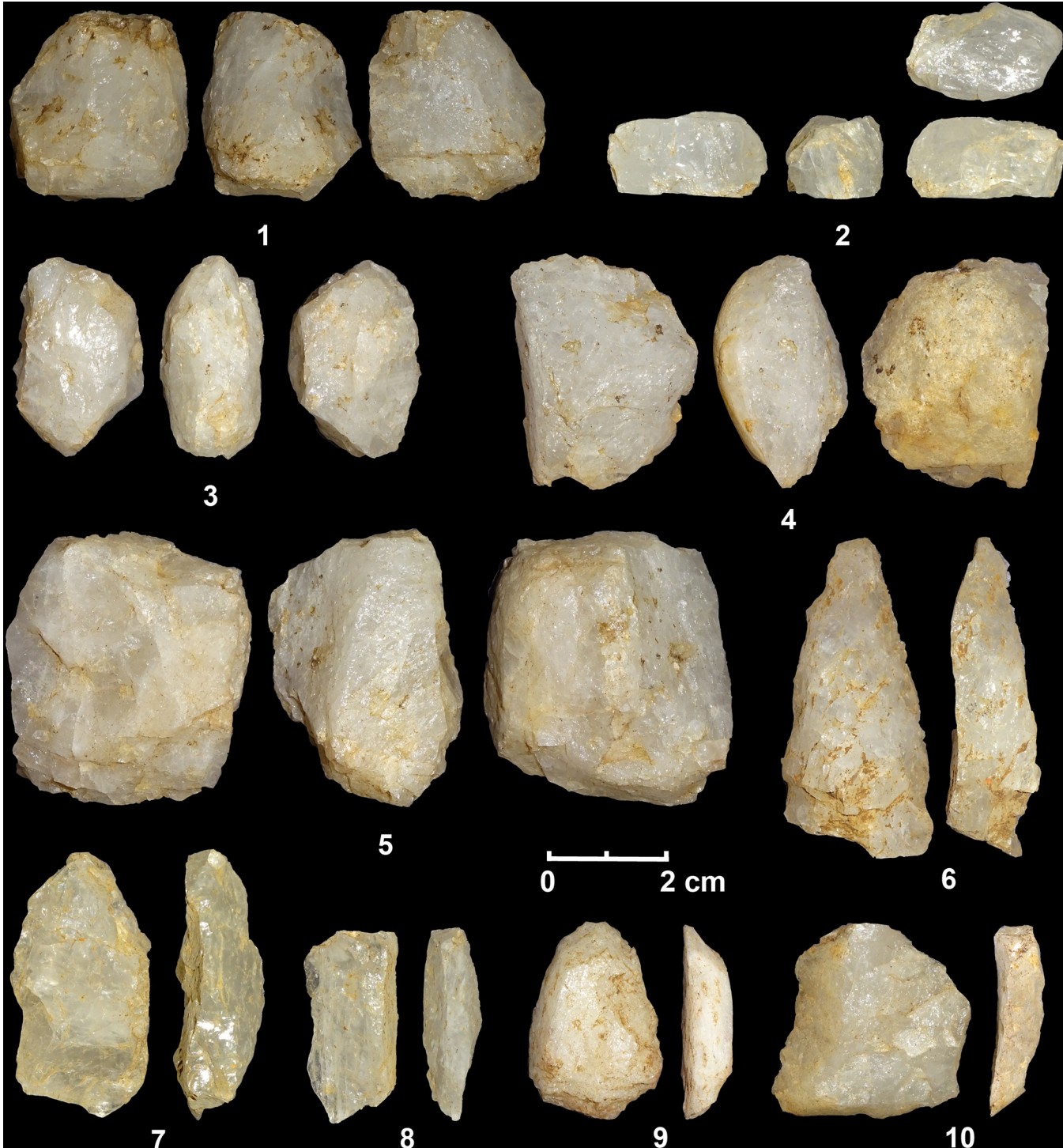

**Fig 9. Bipolar cores (1–5) and flakes (6–1) from the Terminal Pleistocene contexts of Kitulgala Beli-lena.**

produced by freehand percussion. In the second core in milky quartz (weight: 16.2 g; length: 29.5 mm, width: 24 mm, thickness: 18.7 mm), a portion of the blank broke and the fracture was used as a new striking platform for the detachment of two unidirectional flakes by

**Table 8. Counts, mean (μ) and standard deviation (σ) of the metric attribute (mm) of the bipolar-on-anvil cores of Kitulgala Beli-lena in the Terminal Pleistocene phase.**

| | Attributes | Crystal | | | Milky | | |
|---|---|---|---|---|---|---|---|
| | | N° | μ | σ | N° | μ | σ |
| **Vertical** | Weight | 11 | 20 | *12.4* | 28 | 28 | *22.8* |
| | Length | | 29.3 | *5.3* | | 33.5 | *9* |
| | Width | | 26.8 | *5.9* | | 29.2 | *10.1* |
| | Thickness | | 19.3 | *6* | | 20.6 | *7* |
| **Horizontal** | Weight | 4 | 26 | *15.3* | 5 | 47.8 | *35.7* |
| | Length | | 28.5 | *2.3* | | 34.3 | *12.4* |
| | Width | | 33.5 | *10.5* | | 36.4 | *10.3* |
| | Thickness | | 22 | *3.4* | | 24.8 | *7.8* |
| **Orthogonal** | Weight | 1 | 23.4 | | 5 | 19.6 | *11.6* |
| | Length | | 30.8 | | | 30.2 | *3.7* |
| | Width | | 36.4 | | | 25.4 | *3.7* |
| | Thickness | | 21.3 | | | 19.8 | *5.4* |

freehand percussion. Other examples of opportunistic exploitation are: one flake in milky quartz (weight: 30.7 g; length: 40.1 mm, width: 24.1 mm, thickness: 21.5 mm) in which the semi-cortical portion was used as striking platform for the production of one small flake; one cortical pebble (weight: 159 g; length: 56 mm, width: 62.1 mm, thickness: 39.5 mm) that was first split along the longest axis followed by one unidirectional flakes removal from the ventral surface by freehand percussion (Table 7).

Analysis of the flake assemblage documents larger frequencies of items in crystal and milky quartz (Table 7, Fig 9). Cortical flakes and cortical fragments corroborates the import of pebbles at Kitulgala Beli-lena and primary decortication activities at the site (Table 7). Splinter pieces are also recorded in two semi-cortical flakes in crystal quartz, and in one cortical and in four semi-cortical flakes in milky quartz. Chert artefacts are few in number, and by-products of different knapping events, probably transported as components of the toolkit (Table 7). Examination of the direction of removals on the flakes' dorsal side reveals the presence of bidirectional (64%) and unidirectional (35%) patterns while orthogonal removals are found in only one flake in crystal quartz and one in milky quartz. The technological characteristics of the complete flakes support the association with bipolar reduction and only one flake could be related with freehand knapping. Other typical bipolar by-products are splinter flakes, a *bâton-net* flake and siret knapping accidents. These latter blanks are more numerous during the later phases of reduction and siret fractures *sensu strico* are more frequent than siret fractures *sensu lato* (Table 7).

Examination of the frequency of complete flakes by length intervals shows that production is aimed towards blanks smaller than 20 mm, and larger artefacts are few (Fig 8, Tables 4 and 5). Statistical comparison reveals a significant difference in the median values of cortical and complete flakes in crystal quartz (Mann–Whitney test, $p = 0.0050$) and milky quartz (Mann–Whitney test, $p = 0.0003$), and in the median values of cortical flakes (Mann–Whitney test, $p = 0.0110$) and complete flakes (Mann–Whitney test, $p = <0.0001$) in crystal and milky quartz.

## Holocene (10,577–8,029 cal BP)

The lithic assemblage of the Holocene phase includes 3,603 artefacts (3,596 lithic items and 7 hammers) (Tables 9, S1 and S2). The primary raw materials used are crystal and milky quartz

**Table 9. Total number and percentages of lithic artefacts in the Holocene.**

| | Crystal | % | Milky | % | Veiny | % | Rose | % | Grainy | % | Chert | % | Total | % |
|---|---|---|---|---|---|---|---|---|---|---|---|---|---|---|
| Cortical flake >50% | 11 | 0.4 | 12 | 2 | 2 | 5.7 | 1 | 5 | 1 | 1.6 | | | 27 | 0.8 |
| Cortical flake <50% | 34 | 1.2 | 31 | 5.3 | | | | | 1 | 1.6 | | | 66 | 1.8 |
| Flake | 303 | 10.5 | 106 | 18 | | | 3 | 15 | 3 | 4.9 | 3 | 60 | 418 | 11.6 |
| Splinter flake | 4 | 0.1 | 7 | 1.2 | | | | | | | | | 11 | 0.3 |
| Bladelet | 9 | 0.3 | | | | | | | | | | | 9 | 0.3 |
| Cortical fragment | 14 | 0.5 | 25 | 4.3 | 5 | 14.3 | 3 | 15 | | | | | 47 | 1.3 |
| Siret x1 | 4 | 0.1 | 5 | 0.9 | 1 | 2.9 | 1 | 5 | | | | | 11 | 0.3 |
| Siret x2 | 3 | 0.1 | | | | | | | | | | | 3 | 0.1 |
| Siret x3 | 1 | 0.03 | | | | | | | | | | | 1 | 0.03 |
| Fragment | 1810 | 62.7 | 276 | 46.9 | 23 | 65.7 | 6 | 30 | 37 | 60.7 | 1 | 20 | 2153 | 59.9 |
| Siret x1 | 106 | 3.7 | 26 | 4.4 | 2 | 5.7 | 2 | 10 | 1 | 1.6 | 1 | 20 | 138 | 3.8 |
| Siret x2 | 5 | 0.2 | | | | | | | | | | | 5 | 0.1 |
| Siret x3 | 7 | 0.2 | 3 | 0.5 | | | | | | | | | 10 | 0.3 |
| Chips | 534 | 18.5 | 66 | 11.2 | 1 | 2.9 | 2 | 10 | 17 | 27.9 | | | 620 | 17.2 |
| Core | 20 | 0.7 | 22 | 3.7 | | | | | 1 | 1.6 | | | 43 | 1.2 |
| Core fragment | 22 | 0.8 | 9 | 1.5 | 1 | 2.9 | 2 | 10 | | | | | 34 | 0.9 |
| Total | 2887 | 100 | 588 | 100 | 35 | 100 | 20 | 100 | 61 | 100 | 5 | 100 | 3596 | 100 |

with other quartz varieties being knapped in lower percentages (Table 9, Fig 4). Chert artefacts comprise three flakes, one fragment and one siret *sensu stricto* (Table 9). The knapping method most frequently used is bipolar (Fig 10); only two cores were reduced by freehand percussion (Table 10). Ten cores show partial fragmentation of the proximal and/or distal side and are excluded in the descriptive statistics. The modality, vertical axial of the bipolar method, is documented in a higher frequency, followed by orthogonal and horizontal modalities (Table 10). Examination of vertical axial cores shows typical morphologies derived by compression of the artefacts between the hammer and the anvil, with detachments on the proximal and distal sides and removals on the dorsal surfaces with a low portion of cortex (26%). Production is limited to one flaking surface, while the dorsal side remains cortical; this is only documented by one core on milky quartz. Cores exploited until exhaustion total to only two artefacts (crystal = 1, milky = 1), and one core in milky quartz is characterized by a flat distal surface. Comparison between metric attributes of the vertical axial cores in crystal and milky quartz reveal significant differences in weight (Mann-Whitney test, $p = 0.0245$) and in length (Mann-Whitney test, $p = 0.0108$).

The second main reduction strategy used in the assemblage is the orthogonal modality (Table 10). During bipolar knapping, cores were rotated 90° degrees, and at least two striking platforms were used for the production of small blanks. Owing to rotation, the flaking surfaces show the presence of crossed removals, and the cores' outlines are shaped as a quadrangular morphology. A crystal quartz core shows exploitation until exhaustion, while another core shows turning after the fracture of the first striking platform. Descriptive statistics suggest no substantive difference between the weight and size of the artefacts in crystal and milky quartz.

Some bipolar cores show exploitation using the horizontal modality (Table 10). Three cores in crystal quartz are characterized by an axial reduction in which the points of contact hammer/core and core/anvil are vertically aligned. Conversely, the points of contacts of the core in milky quartz are off-axis, shaping the blank into a pyramidal morphology and producing elongated blanks.

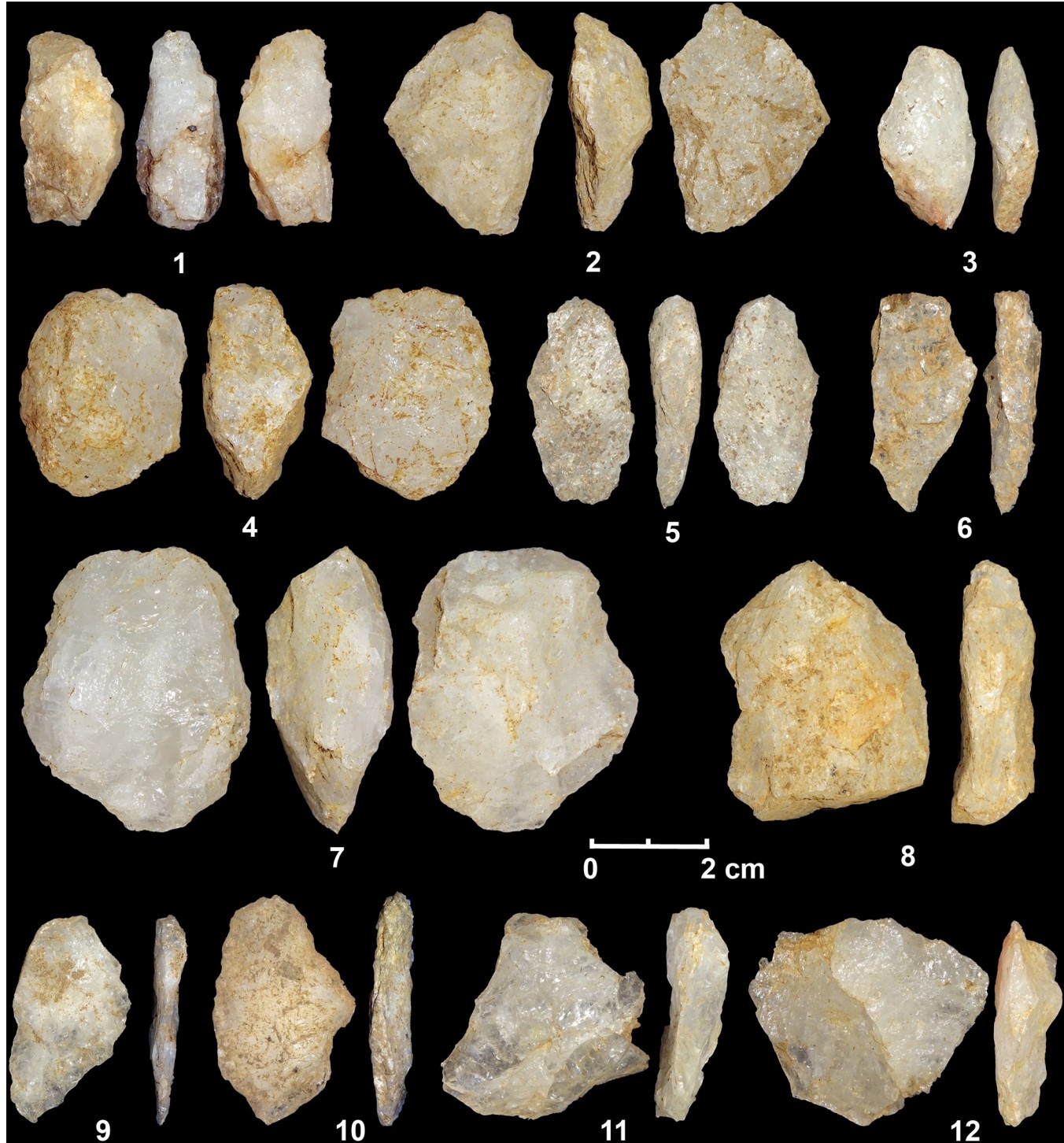

**Fig 10. Bipolar cores (1–5, 7) and flakes (6, 8, 9–12) from the Holocene contexts of Kitulgala Beli-lena.**

Other types of cores are also found in the assemblage (Table 10). A bipolar core fragment (vertical axial) in milky quartz was recycled for a short knapping sequence by freehand, and the production of three small flakes (Table 10). Analysis reveals one unidirectional core in

**Table 10. Total number and percentages of the types of cores by different raw materials in the Holocene phase.**

|  | Crystal | % | Milky | % | Grainy | % | Total | % |
|---|---|---|---|---|---|---|---|---|
| **Vertical** | 11 | *55* | 17 | *77.3* |  |  | 28 | *65.1* |
| **Horizontal** | 3 | *15* |  |  |  |  | 3 | *7* |
| **Horizontal non-axial** |  |  | 1 | *4.5* |  |  | 1 | *2.3* |
| **Orthogonal** | 5 | *25* | 3 | *13.6* |  |  | 8 | *18.6* |
| **Bipolar + unid.** |  |  | 1 | *4.5* |  |  | 1 | *2.3* |
| **Unidirectional** | 1 | *5* |  |  | 1 | *100* | 2 | *4.7* |
| **Total** | 20 | *100* | 22 | *100.0* | 1 | *100* | 43 | *100* |

crystal quartz (weight: 38.8 g; length: 29.7 mm, width: 43.4 mm, thickness: 25.5 mm) and one small pebble in grainy quartz (weight: 8.2 g; length: 22.8 mm, width: 17.6.4 mm, thickness: 21.1 mm) that present three removals, probably produced by freehand or anvil-rested action.

Comparison of metric attributes indicates no significant differences between the knapping modalities of bipolar cores (Table 11). However, the distribution of length and width values presents a broader variability, suggesting different degrees of artefact reduction (Fig 6). Orthogonal cores show a decreasing trend in size as reduction continues whereas vertical and horizontal cores reveal different lengths while maintaining similar width values (Fig 6). This pattern is also evident in the linear regression model although the difference between the slopes is not significant (F = 0.882869, DFn = 2, Dfd = 24, *p* = 0.4266) (Fig 7).

Analysis of the flake assemblage indicates that phases of decortication and production were carried out at the site. Cortical and semi-cortical flakes are more numerous in crystal and milky quartz and four splintered pieces retain a small portion of cortex on their dorsal side (Table 9). Technological study indicates that flakes were produced with the bipolar method (Fig 10) and blanks that could potentially be related with freehand were not present. Although the bipolar orthogonal modality is common in the assemblage, only two orthogonal flakes in crystal quartz were identified. The main patterns of scar removals on the dorsal side of complete flakes are unidirectional (51.4%) and bidirectional (48.3%). These unbroken flakes display diffuse bulbs, battered platforms, and fragmented distal sides. Other observed by-products typical of bipolar knapping are splintered pieces, and siret knapping accidents, in particular the category siret *sensu stricto* (Table 9).

**Table 11. Counts, mean (μ) and standard deviation (σ) of the metric attribute (mm) of the bipolar-on-anvil cores of Kitulgala Beli-lena in the Terminal Pleistocene phase.**

| Phase | Attributes | Crystal | | | Milky | | |
|---|---|---|---|---|---|---|---|
|  |  | N° | μ | σ | N° | μ | σ |
| Vertical | Weight | 4 | 6.6 | *1.9* | 15 | 21.5 | *17.5* |
|  | Length |  | 20.7 | *2.8* |  | 33.6 | *7.1* |
|  | Width |  | 21.2 | *3.7* |  | 28.4 | *9.7* |
|  | Thickness |  | 13.3 | *2.9* |  | 18.5 | *5.9* |
| Horizontal | Weight | 3 | 10.4 | *7.8* | 1 | 18.9 |  |
|  | Length |  | 25.8 | *6.7* |  | 25 |  |
|  | Width |  | 26.6 | *9.3* |  | 27.1 |  |
|  | Thickness |  | 13.1 | *3.4* |  | 19 |  |
| Orthogonal | Weight | 5 | 23.1 | *8.5* | 3 | 24.9 | *19.9* |
|  | Length |  | 33.0 | *3.3* |  | 29.1 | *7.7* |
|  | Width |  | 31.9 | *8.9* |  | 28.6 | *12* |
|  | Thickness |  | 20.7 | *2.8* |  | 18.2 | *4.6* |

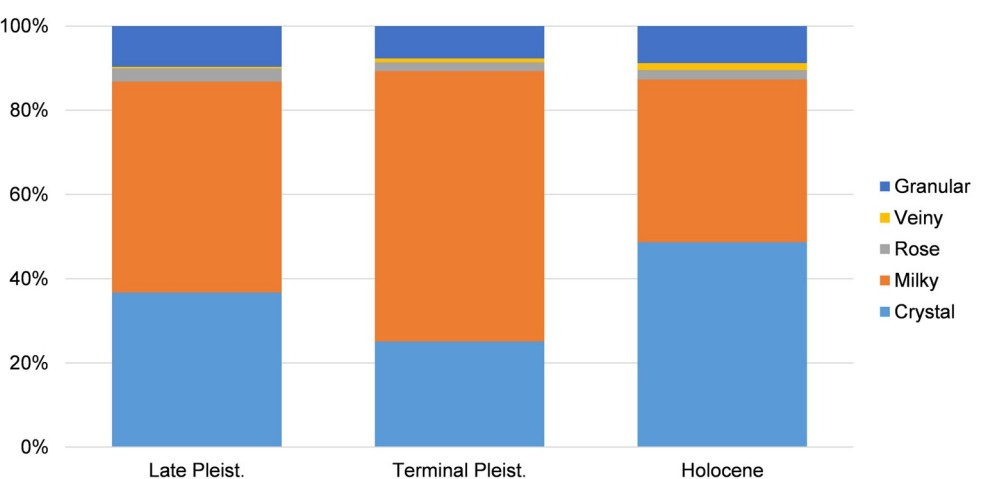

**Fig 11. Frequency of the weight of the quartz type by chronological phases recorded at Kitalgala Beli-lena.**

Comparison of the frequency of complete flakes by length intervals shows that most of the blanks are smaller than 20 mm whereas the items falling within the interval of 30 and 40 mm are documented in lower percentages (Fig 8, Tables 4 and 5). Statistical comparison reveals a significant difference in the median values of cortical and complete flake in crystal quartz (Mann–Whitney test, $p = 0.0083$), between cortical flakes (Mann–Whitney test, $p = 0.0182$), and between complete flakes (Mann–Whitney test, $p = <0.0001$) in crystal and milky quartz.

## Summary

Analysis of the Kitulgala Beli-lena lithic assemblage reveals long-term technological stability from *c.* 45,000 to 8,000 years BP (Tables 1–11, Figs 8 and 11). Even if a hiatus is present at the site between 31 and 17,000 cal BP [34], climate change at the end of MIS 3 and the Last Glacial Maximum do not appear to have driven the development of new technical behaviours. The bipolar method was chosen for the production of small tools throughout occupation of the site (Figs 6 and 8). Technological continuity from the Late Pleistocene to the Holocene is reinforced by the maintenance of certain technically expedient practices during the knapping processes. Siret-type fractures could occur during the cores' reduction but the patterns of rotating the core and using the flat surface of the fracture for a more stable placing on the anvil is an approach found only at Kitulgala Beli-lena. This approach is absent, for example, at nearby Fa-Hien Lena [38]. From the Late Pleistocene onwards, the frequency of crystal quartz increases as the other quartz varieties decrease substantially (Fig 4). However, comparison of weight percentages indicates larger values for milky quartz, especially during the Terminal Pleistocene (Fig 11). This result may be related to higher fragmentation rates of nodules in crystal quartz, whereas the reduction of pebbles in milky quartz produced fewer, but larger and heavier blanks. This pattern is documented in all of the chronological phases among cores, and also among complete cortical and non-cortical flakes (Tables 3–5, 8 and 11). Comparison of technical behaviours shows that the quartz pebbles were reduced mostly on their longer axis, while recurrent rotation of the cores is documented in similar frequencies throughout the sequence (Tables 2, 7 and 10). Although backed microliths have not been recovered, the occurrence of flakes shorter than 20 mm supports the microlithic character of the assemblage (Fig 8). This pattern could not be considered accidental owing to the reduction strategy employed, since the bipolar method could also be used to produce larger by-products (e.g. [104,135]).

Furthermore, the main raw material, quartz, can be knapped using a broad array of technologies [102,136,137]. The continuous use of the bipolar knapping method, without variations for millennia, underlines the significant efficacy of this technical behaviour in the subsistence strategies in the rainforest habitat of Sri Lanka.

## Discussion

The expansion of *Homo sapiens* out of Africa from MIS 6 onwards [5,9,138–141] has been equated with the development of projectile technologies [142–144], enabling the hunting of a broad variety of prey while adapting to a wide-range of environments. From ~60,000 years ago, microlithization was clearly an additional component of the technical package used by *Homo sapiens* during the dispersal of populations across Eurasia [112,113,145]. Together with laminar/lamellar technologies, the bipolar method was broadly used in different ecological habitats given its ability to consistently generate straight blanks suitable for hafting with minimal modification of the edges [124]. However, under the bipolar umbrella, regional differences have been documented in the use and the modification of by-products (e.g. Uluzzian [146]). Shedding light on this variability in different environmental contexts is particularly pivotal for understanding the foraging success of *Homo sapiens* in some of the more 'extreme' habitats it came into contact with during the Late Pleistocene. Analysis of the lithic assemblages from Kitulgala Beli-lena indicates long-term technological continuity in the tropical rainforest of Sri Lanka from *c*. 45,000 years ago BP to the Holocene. This evidence suggests that, since the onset of settlement of the Sri Lankan tropical rainforest by *Homo sapiens*, the combined use of the bipolar method with bone tool technology allowed successful, resilient resource exploitation in this part of South Asia [32–34,37,38].

Tropical rainforests tend to have high primary biomass and a broad range of medium- and small-sized animals, and edible fruits (e.g. breadfruit) are available year-round if technological strategies are developed to exploit them. However, due to the high-density of vegetation cover and the elusive nature of prey, foragers may need to move frequently [42,82]. The distance covered between relocations could be small, however, in contrast to examples in more seasonal habitats. For example, in Malaysia, 20[th] century observations indicated that the Semang relocate their residential camp 26 times a year trekking ~11 km; in contrast, the Penan perform 45 moves per year covering a distance of only ~8 km [42,49]. The arrival of the monsoon and months of intense rains could seasonally influence the route and frequencies of these displacements in different parts of Asia. Present day Mani hunter-gatherers in Thailand, for instance, retreat in caves during the rainy periods, while they prefer to live in temporary encampments in the cool rainforest during the dry season [147,148]. Thus, during the monsoon period, daily forays from the camp or moves between different natural shelters could occur [147]. Generally, tropical foragers acquire a large proportion of their calories from meat and in lesser percentages from fish or plants/fruits [42]. Since small animals can hide in the dense vegetation, foraging activities require planning ahead, with toolkit production in advance of anticipated use [42,49,52]. From this perspective, light and portable toolkits, characterized by generalized tools and organic materials (e.g. bamboo, bones) are preferred and useful in seeking opportunistic game (e.g. residential moves [42]).

In lithic studies, technological organization and the patterns of raw material acquisition are used as general proxies for estimating the mobility of prehistoric hunter-gatherers [149,150]. Systematic mapping of raw material distributions over the Sri Lankan' wet zone is unfortunately still missing. However, preliminary surveys indicate that quartz pebbles are readily found in nearby streams and in open sedimentary sections, in the vicinity of Kitulgala Beli-lena. Examination of the lithic assemblage reveals diachronic changes in raw material

procurement (Fig 4). From the Late Pleistocene, the frequency of crystal quartz increases whereas other quartz varieties decrease substantially (Fig 4). This pattern may be related to changes in the availability of quartz varieties in the stream or a shift in the areas of procurement. Ethnographic data indicate that rainforest foragers move their residential base frequently [42,151]. Thus, it is likely that through time, hunter-gatherers exploited different ecological zones in the proximity of Kitulgala Beli-lena. During these forays, quartz pebbles from other outcrops could have been gathered following embedded raw-material procurement strategies. However, comparison of the number of the cores shows a greater quantity of milky quartz artefacts during the Late and Terminal Pleistocene (Tables 2 and 7), whereas the number of complete flakes and fragments is more numerous in crystal quartz (Tables 1 and 6). Generally, crystal quartz fractures more easily than other varieties, and this discrepancy between the number of cores and flakes in the milky quartz assemblage could be related to an export of the by-products off-site. Other information on the patterns of toolkit transport may be drawn from the chert artefacts. These items comprise simple flakes, fragments and one core-on-flake, recovered in the Terminal Pleistocene phase (Tables 1, 6 and 9). These chert artefacts were not curated and comprise knapping by-products and blanks, which are generally interpreted as waste in lithic production with little utility value (Tables 1, 6 and 9). However, they reflect events of knapping activities carried out in the rainforest and imported to the rockshelter.

Study of the Kitulgala Beli-lena lithic assemblage reveals the absence of curated or multifunctional stone artefacts (e.g. scrapers, points) (Tables 1, 6 and 9). The bulk of the toolkit is composed of quartz flakes and fragments (Tables 1, 6 and 9). The use of retouch for reshaping or modifying the shape of the blanks is also lacking. Although these quartz microliths could be inserted into sophisticated composite projectiles with bone points [30,33,38,41], the gear for accomplishing extractive tasks is apparently restricted to small unmodified blanks. The demand of fresh cutting edges for coping with domestic activities (e.g. carcass butchering) was thus seemingly met by the import of quartz pebbles to be reduced on site. In previous fieldwork at the site, only 27 backed microlithic were found [35], a number that is extremely low in comparison to the tens of thousands of quartz lithic items discovered. A similar pattern was also observed at Fa-Hien Lena [38] and Batadomba-lena [36] where the number of backed microliths is extremely limited and retouched tools absent. These data indicate that the production of retouched stone tools or the resharpening of the quartz flakes was unnecessary for the exploitation of the tropical environment.

At Kitulgala Beli-lena, the high-density of lithic materials and the complete core reduction sequences *on site* are in accordance with the recognized criteria for interpreting long-term occupations [152]. Isotopic data on human remains from Batadomba-lena, Fa-Hien Lena and Balangoda Kuragala indicate that forest resources were exploited year-round [32]. Given the similarity of the archaeological remains found at Kitulgala Beli-lena to these sites, it is likely that this site was also one of the main residential bases of *Homo sapiens* living in the Sri Lankan rainforest.

Although these caves and rockshelters could have been more intensively occupied during the rainy seasons, the occupational redundancy at Kitulgala Beli-lena and the other sites in the wet zone shows behaviours that do not align completely with ethnographic observations. Generally, foragers in contemporary rainforests are highly mobile and frequently move their home base during the year. The Punam of Borneo or the Guayaki of Paraguay will not even relocate their camps closer to previously used locations [51,153,154]. This comportment makes their residential sites barely visible and very ephemeral [51]. Conversely, in Sri Lanka, the abundance of archaeological materials and the repeated use of the caves and rockshelters [35,36,38,39,44] is a behaviour closer to the model proposed by Binford in which foragers in a

high biomass environment tend to settle in foraging zones that allow frequent logistical forays and few residential moves per year. From another perspective, the prolonged settlement is also similar to the examples of tethered nomadism, in which foragers are dependent on particular resources or features of the landscape, and occupy such places for a long period of time (e.g. water sources in deserts or highly seasonal environments) [155]. Even if the presence of a stream close to the sites would have provided access to clean water and freshwater resources all year long, it is more likely that in the locations of the caves and rockshelters most of the resources had a degree of reproducibility that could withstand relatively high rates of consumption without being depleted.

Animal bone remains in the 2017 excavations at Kitulgala Beli-lena were found only in the early Holocene contexts [34] whereas in other areas of the rockshelter, they were documented throughout the sequence [39,108]. The most parsimonious explanation for this suggests either a difference in soil acidification or that prey butchering and consumptions was performed in different locations of the site. Overall, the current data indicate that arboreal and semi-arboreal species were preferred [34]. Comparison of this assemblage with the faunal variety found at Fa-Hien Lena [33] and Batadomba-lena [31,32] indicates strong similarities implying that analogous types of prey were hunted during the Late and the Terminal Pleistocene, and Holocene. As in the other cave sites of the wet zone, hunting activities were complemented by the gathering of molluscs and fruits (e.g. wild breadfruit, kekuna nut) [34] and by the use of snares and traps [30].

Recent functional studies point out that small quartz fragments could be inserted unretouched into composite projectiles and therefore play an important role in hunting strategies [156]. Thus far, microwear analysis has not been performed on the collection of Kitulgala Beli-lena. However, the abundance of microliths, and the discovery of some of the earliest human use of bow and arrow technology at Fa-Hien Lena by ~48,000 years ago in the form of bone points [30,33] corroborates the hypothesis that the lithic production was aimed to produce elements for hunting weapons.

All of these data highlight that the persistent utilization of the bipolar method at Kitulgala Beli-lena, and Sri Lanka more widely, was key to the adaptation of early groups that modified their technologies to respond to the unique environment of the rainforest zone. From this perspective, the Kitulgala toolkits should not be interpreted as having a lower degree of complexity due to the reduced number of components [58] or in the absence of curated stone-tools [52]. Instead, the toolkit should be understood as result of a process of technical selection in which the bipolar method was chosen among a diverse array of technological choices. As *Homo sapiens* dispersed across South Asia prepared-core or laminar technologies that were deployed in more seasonal habitats [13,157], were abandoned in favour of the use of the bipolar method in tropical, humid forests. This transition was probably driven by the higher benefits that these simpler lithic items provided in the exploitation of a high biomass environment. This alternative viewpoint would avoid interpretation of unstandardized artefacts in the rainforest as a cultural issue, and instead include them in a broader group, together with bone point projectile technologies, as part of diverse, sophisticated foraging strategies. In comparison with other tropical areas of South-East Asia (e.g. Indochina, Borneo, East Timor, Philippines), Sri Lanka reveals similar trajectories of adaptation to the rainforest, comprising: 1) the maintenance of expedient/amorphous stone tools across time, 2) the development of bone projectile technologies, 3) the hunting of semi arboreal/arboreal species, and 4) the use of a variety of plants. Within this foraging behaviour, regional variabilities are, however, recorded. In Indochina, the Hoabinhian is characterized by simple flake-core technology, choppers and unifacial flaked pebble tools, called Sumatraliths [158–162]. In Borneo, the lithic assemblages are categorised by the production of simple freehand cores, flakes, and choppers with very few retouched artefacts [163,164]. In East Timor, the bipolar assemblage is supplemented by core-

on-flakes, truncated faceted pieces and simple cores [165] whereas in the Philippines and Indonesia non formal lithic artefacts are common [166–168].

From a broader perspective, the process of adaptation to various rainforests settings by *Homo sapiens* in the Late Pleistocene often seems to have involved the selection of low-cost technologies for achieving a higher foraging efficiency. Carbonell and colleagues [169] postulated a progressive development of early technical systems in which the loss of homogeneity in lithic production drove the consolidation of variability and diversity. In Late Pleistocene tropical forests, the processes of stabilization of low-cost lithic behaviours seems to follow a different trajectory. Despite the diversity of knapping technologies used by *Homo sapiens* populations during their expansion into South Asia and South-East Asia, there is an apparent convergence towards the use of expedient artefacts. This broad technical homogeneity subsequently consolidated into different regional toolkits. In local contexts, the establishment of the rainforest "package" may have followed different paths owing to the interplay between demographic and cultural dynamics and variations in the availability of alternative resources through time (e.g. wood, bamboo) [78,166].

Study of the lithic assemblages of Kitulgala Beli-lena shows the recurrent use of the bipolar technique, indicating technological stability spanning from the Late Pleistocene to the Holocene. Even if the utilization of expedient tools is often interpreted to be the result of a lower degree of behavioural complexity, our data from the Pleistocene and Holocene contexts of Kitulgala Beli-lena point out the successful exploitation of the rainforest since *c.* 45,000 years BP. Similar approaches to lithic production were also documented at Fa-Hien Lena [38] and Batadomba-lena [48]. The combination of bipolar by-products with osseous projectile points and the targeting of semi-arboreal/arboreal species was part of a wide-ranging, specialized foraging strategy that allowed the widespread and recurrent settlement of the wet zone of Sri Lanka.

## Supporting information

**S1 Fig. Diacritical schemes of bipolar cores (1–7) and flakes (8–12) from the Late Pleistocene contexts of Kitulgala Beli-lena.**
(PDF)

**S2 Fig. Picture and diacritical scheme of the bipolar anvil-rested from context 19 of Kitulgala Beli-lena.**
(PDF)

**S3 Fig. Diacritical schemes of bipolar cores (1–5) and flakes (6–1) from the Terminal Pleistocene contexts of Kitulgala Beli-lena.**
(PDF)

**S4 Fig. Diacritical schemes of bipolar cores (1–5, 7) and flakes (6, 8, 9–12) from the Holocene contexts of Kitulgala Beli-lena.**
(PDF)

**S5 Fig. Histogram of the frequency of complete flakes by width intervals during the different chronological phases at Kitulgala Beli-lena.**
(PDF)

**S1 Table. Total number of lithic artefacts by chronological phase at Kitulgala Beli-lena.**
(PDF)

**S2 Table. Total number of cores by chronological phase at Kitulgala Beli-lena.** Categories: Bipolar Vertical; Bipolar Horizontal; Bipolar Horizontal non-axial; Bipolar orthogonal; Bipolar

anvil-rested; Bipolar + free hand unidirectional; Free hand unidirectional.
(PDF)

**S3 Table. P values of Shapiro-Wilk normality test (alpha = 0.05) from cores and unbroken flakes by chronological phase at Kitulgala Beli-lena.** Values in bold are normally distributed.
(PDF)

**S1 File.**
(XLSX)

## Acknowledgments

We dedicate this work to the memory of our co-author Prof. Siran Deraniyagala who recently passed away. We acknowledge Prof. S. Dissanayake, Prof. P.B. Mandawala, Mr. S.A.T.G. Priyantha, and other members at the Excavations Branch of the Department of Archaeology, Sri Lanka, for assistance and the support of our fieldwork. We thank Mr L.V.A. De Mel, Mr. K. K. Ruwan Pramod, Mr. P.G. Gunadasa, Mr Suranga Jayasinha, Mr. K.A.S Lakmal, Mr. Tiran Ananda and Mr. J. Perera for their participation in the fieldwork. For support, we thank Prof. Alexander Kapukotuwa and other faculty members of the Department of History and Archaeology of the University of Sri Jayewardenepura. We appreciate the cooperation of Mr. Sunil Kannangara (District Secretary of Colombo).

## Author Contributions

**Conceptualization:** Andrea Picin, Oshan Wedage, Nicole Boivin, Patrick Roberts, Michael Petraglia.

**Data curation:** Andrea Picin, Oshan Wedage.

**Formal analysis:** Andrea Picin, Oshan Wedage, James Blinkhorn, Noel Amano.

**Funding acquisition:** Nicole Boivin, Michael Petraglia.

**Investigation:** Andrea Picin, Oshan Wedage, James Blinkhorn, Noel Amano, Siran Deraniyagala, Patrick Roberts, Michael Petraglia.

**Methodology:** Andrea Picin, Oshan Wedage, James Blinkhorn, Patrick Roberts, Michael Petraglia.

**Project administration:** Oshan Wedage, Nicole Boivin, Michael Petraglia.

**Resources:** Nicole Boivin, Michael Petraglia.

**Supervision:** Siran Deraniyagala, Nicole Boivin, Patrick Roberts, Michael Petraglia.

**Visualization:** Noel Amano.

**Writing – original draft:** Andrea Picin, Oshan Wedage.

**Writing – review & editing:** Andrea Picin, James Blinkhorn, Noel Amano, Nicole Boivin, Patrick Roberts, Michael Petraglia.

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
