## [Decision Letter · Decision Letter 0]

25 Apr 2022

PONE-D-22-05404Homo sapiens lithic technology and microlithization in the South Asian rainforest at Kitulgala Beli-lena (c. 45 – 8 ka)PLOS ONE

Dear Dr. Picin,

Thank you for submitting your manuscript to PLOS ONE. After careful consideration, we feel that it has merit but does not fully meet PLOS ONE’s publication criteria as it currently stands. Therefore, we invite you to submit a revised version of the manuscript that addresses the points raised during the review process.

Both of the reviewers offer constructive criticism, with the comments of Reviewer 2 being particularly detailed and suggesting some references and asking for some data sharing/clarifications. 

We look forward to receiving your revised manuscript.

Kind regards,

Radu Iovita

Academic Editor

PLOS ONE

Journal Requirements:

3. Please amend the manuscript submission data (via Edit Submission) to include author Siran Deraniyagala.

4. We note that Figure 1 in your submission contain map images which may be copyrighted. All PLOS content is published under the Creative Commons Attribution License (CC BY 4.0), which means that the manuscript, images, and Supporting Information files will be freely available online, and any third party is permitted to access, download, copy, distribute, and use these materials in any way, even commercially, with proper attribution. For these reasons, we cannot publish previously copyrighted maps or satellite images created using proprietary data, such as Google software (Google Maps, Street View, and Earth). For more information, see our copyright guidelines: http://journals.plos.org/plosone/s/licenses-and-copyright.

Reviewers' comments:

Reviewer's Responses to Questions

**Comments to the Author**

1. Is the manuscript technically sound, and do the data support the conclusions?

Reviewer #1: Yes

Reviewer #2: Yes

2. Has the statistical analysis been performed appropriately and rigorously? 

Reviewer #1: Yes

Reviewer #2: Yes

3. Have the authors made all data underlying the findings in their manuscript fully available?

Reviewer #1: Yes

Reviewer #2: No

4. Is the manuscript presented in an intelligible fashion and written in standard English?

Reviewer #1: Yes

Reviewer #2: Yes

5. Review Comments to the Author

Reviewer #1: This interesting paper describes the lithic assemblages discovered at Kitulgala Beli-Iena, compares them with stone tools found at other sites from the region, and links the findings to other components of the economy of hunter-gatherers living at the sites and in Sri Lanka.

The authors managed to relate their findings to broader concepts such as the relationship between lithic technology and mobility or resource availability, making the reading of the article rather enthralling and thought-provoking.

I would suggest a few additions and the correction of what appears to be a typo:

1) Introduction section

During the early to Late Pleistocene, hominins reached new geographical areas as they expanded out of Africa, populations eventually extending "FROM" the far western [1-5] and eastern edges of Eurasia [6-9].

I believe that the authors meant “to” the edges of Eurasia

2) Materials and methods section

I would suggest adding some schematic drawings to explain visually the terminology used in this study to describe the different knapping methods.

3) Discussion section

-I would suggest adding to the discussion a comparison between the lithic industries from Kitulgala Bali Iena and other sites from Sri Lanka, in particular the question of presence/absence of standardised geometric microliths. These are absent from the assemblages reported in this paper, unlike at other sites from Sri Lanka such as Batadomba-Iena. How can this be explained? What could this mean?

4) Bone points: as the complementarity between lithic and bone tools is mentioned at several places in the manuscript and forms an important idea in the discussion, I would suggest adding a figure showing some of these artefacts.

5) Eventually, as part of the Open Data scheme of the journal, I would suggest adding tables with all the measurements of the individual artefacts and possibly some diacritic diagrams of the knapping techniques identified as Supplementary Material.

Reviewer #2: This is a very interesting paper on the Late Pleistocene-Holocene lithic assemblage of Kitulgala Beli-lena. The corpus of data is robust, the paper is well written and the discussion is interesting. The paper is mainly descriptive but there is not a problem with that.

My apologies for my poor English writing. If there is something not clear on my comments I will be happy to clarify via email. Moreover, I would like to apologize for sending my review one week late, I know as an author how frustrating is to be waiting to the reviewers.

I have some remarks and suggestions, mainly to strengthen this manuscript.

• I have some concerns regarding the stratigraphical research units (divisions) analysed. The description of the stratigraphy is very short, the stratigraphy shown in Figure 1 is quite simple and the decision of merging many layers as units of study for the lithics should be explained and justify. Figure 1 shows in the right-hand size a stratigraphy. The Late Pleistocene seems to have several layers. A better description of the stratigraphy and the decision of merging the lithic analysis in three big chronological phases should be justified and properly explained. First, I would dedicate some part of the introduction on the purely description of the stratigraphy and later on, in the methodology/sample section, I would explain why the authors use such big units. Using these three big chronological groups (Late Pleistocene, Terminal Pleistocene and Holocene) might be masking some diachronic variations (which would change completely the discussion of this paper). Maybe not, but then it must be justify. Moreover, I do not know if I missed it but I would specify what is the surface excavated and the relationship of litres of sediment/artefacts.

• Since the beginning of the manuscript, they use the term ‘microlithism’. In the methodology section they explain that they follow Shea and Pargeter definition. That clarification is pertinent as there is a lot of confusion regarding that term in lithic analyses, but I would explain this before in the manuscript. Because when I started reading, I was wondering what definition are the authors following and I wrote it in my notes. On this topic, in the last paragraph of the Summary, there is no need to mention that there are not backed pieces.

• Regarding the quartz variants, as it is the main raw material I wonder if the authors could differentiate varieties within the groups, as for example suggested by: Martínez Cortizas, A. and C. Llana Rodríguez (1996) Morphostructural variables and the analysis of their effect on quartz blank charecteristics. In: Non-Flint Stone Tools and the Palaeolithic Occupation of the Iberian Peninsula, edited by N. Moloney, L. Raposo and M. Santonja. Pp. 49-53. Tempus Reparatum, 649 Oxford. I have worked quite a bit with quartz and even if quartz is quartz for a geologist, for a stone toolmaker there are very different quality of quartzs depending if it is grainy, if it has flaws, etc which affect a lot on what you can make or not with the raw material. I am not proposing that the authors re-do the analysis, but again, I would explain the categories chosen / divisions made regarding raw material for their analysis. In addition, maybe for future analysis this is an interesting suggestion.

• “We first tested the normality of the data using Shapiro- Wilk tests, and, then, we employed a non-parametric test” I suppose this is because all the data were not normal distributions, but where is the table showing these results? This should be provide somewhere in the paper (Supplementary material or in a table). In this regard, I wonder if the authors are sharing their whole database. When I have published in PlosOne I am always encouraged to share my data and I think the authors should do it.

• “Following the demonstration of the bimodal distribution of blade sizes in southern India [113], we employ a 40mm size threshold and describe flakes, blades (bladelets), and retouched tools smaller than 40mm as microlithic (see [38])” I suppose it is okay to compare with other data, but I would rather show the distribution in length and width of the different blanks (maybe with simple histograms) and asses if the distribution is continuous or there are ‘breaks’ on it. This seems particularly relevant being so homogeneous all the groups selected.

• The anvil rested modality I think it was firstly defined by Callahan. I think it should be referenced (the work I am referring is a special monograph on the Neolithic of Sweden, If I remember correctly) In one of their groups they recognize this type of knapping, it would be convenient to specify how they recognize this, what are the stigma? Also, I would add some photos, etc I have been working on bipolar knapping for a while and I do not think the identification of this type of knapping/variant is straightforward, there is very little experimentation. I ask because I am curious and I am working on it indeed.

• A figure explaining the different types of cores is needed the variants vertical, horizontal, bipolar+unid., unidirectional, etc. In lithic technology every analyst use slightly different categories and for the shake of clarity and future comparisons I recommend this is better explained. In the same vein, what do the authors mean by ‘core-edge’ flakes? Also, regarding ‘debris’ I think I have not seen an explanation of how the authors define this category (well, where they put the measure cut to consider a flake chips or debris).

• I also have a problem with the use of the term ’splintered pieces’ , particularly in an assemblage with a lot of bipolar knapping. It seems the authors are referring to a type of blank (which is not its original definition which was typological). The authors need to give clarity in this regard too. Splitered pieces or pieces esquillees was originally a typological category defined by Sonneville-Bordes, later on was problematic, thus they eliminated it from the type-list of the Upper Palaeolithic because it was considered an a posteiori tool…It is problematic since long time ago but researchers keep using it I do not know why. I would dedicate some explanation too to the different blanks considered in this analysis from bipolar knapping: what is a flake-splinter? How is that category different to what they call splintered flake? I would show all this in methodology, particularly because I Imagine in the future they will do a comparative analysis with other sites in Sri Lanka and it is important to explain all these categories better. I cannot remember if these are the categories used in : Mourre V. Les Industries en Quartz au Paléolithique. Terminologie, Méthodologie et Technologie. Paléo. 1996;8:205-23. Regarding this publication I think to reference it is interesting but the authors must bear in mind that the type of knapping defined by Mourre in that work is to flake big blanks, whereas the type of bipolar knapping they have is towards the production of small things, which is slightly different. To distinguish variants within bipolar knapping is the future as finally researchers are opened to identify it.

• Regarding the comparison of the three groups of lithics analysed I would try to do some comparative statistical test beyond figure 5 and 9, although these two are quite eloquent.

• Finally, how did they differ between crystal and milky bipolar cores when the cores were very reduced/exhausted? By the presence of facets? Translucency? I am curious as I have this problem often.

Finally, I would like to say that I enjoyed the reading a learning from this fascinating assemblage.

Paloma de la Peña

6. PLOS authors have the option to publish the peer review history of their article (what does this mean?). If published, this will include your full peer review and any attached files.

Reviewer #1: No

Reviewer #2: **Yes: **Paloma de la Peña

---

## [Author Response · Author response to Decision Letter 0]

13 Jul 2022

Response to Reviewers

We wish to thank the Editor and the reviewers for their helpful and constructive comments. We acknowledge and accept the suggestions made by reviewers 1 and 2, and feel that these have helped to make our paper stronger. We provide a detailed list of responses to the comments and edits made by the Editor and the reviewers in what follows.

Editor comments

Thank you for submitting your manuscript to PLOS ONE. After careful consideration, we feel that it has merit but does not fully meet PLOS ONE’s publication criteria as it currently stands. Therefore, we invite you to submit a revised version of the manuscript that addresses the points raised during the review process. Both of the reviewers offer constructive criticism, with the comments of Reviewer 2 being particularly detailed and suggesting some references and asking for some data sharing/clarifications.

We thank the Editor for their careful reading of the manuscript, as well as the reviewers for their comments that we feel have helped to make our paper still stronger. We have, as indicated below on a point-by-point basis, addressed their series of specific comments.

1. Please ensure that your manuscript meets PLOS ONE’s style requirements, including those for file naming.

We have corrected the manuscript following PLOS ONE’s style templates.

We have now double-checked that the text of ‘Funding Information’ is correct.

3. Please amend the manuscript submission data (via Edit Submission) to include author Siran Deraniyagala

We have now included the author Siran Deraniyagala in the manuscript submission data.

4. We note that Figure 1 in your submission contain map images which may be copyrighted.

We now provide a new Figure 1 that fulfills the Creative Commons Attribution License criteria (CC BY 4.0).

Reviewer #1

This interesting paper describes the lithic assemblages discovered at Kitulgala Beli-Iena, compares them with stone tools found at other sites from the region, and links the findings to other components of the economy of hunter-gatherers living at the sites and in Sri Lanka. The authors managed to relate their findings to broader concepts such as the relationship between lithic technology and mobility or resource availability, making the reading of the article rather enthralling and thought-provoking.

We thank Reviewer#1 for their kind words about our paper, we are very pleased that they found our paper, “interesting”, “enthralling and thought-provoking”.

I would suggest a few additions and the correction of what appears to be a typo:

1) Introduction section: During the early to Late Pleistocene, hominins reached new geographical areas as they expanded out of Africa, populations eventually extending "FROM" the far western [1-5] and eastern edges of Eurasia [6-9]. I believe that the authors meant “to” the edges of Eurasia.

We thank the reviewer for spotting this error. We have corrected the mistake.

2) Materials and methods section: I would suggest adding some schematic drawings to explain visually the terminology used in this study to describe the different knapping methods.

We have provided a new Figure 3 showing the categories of different types of cores used in the analysis.

3) Discussion section, I would suggest adding to the discussion a comparison between the lithic industries from Kitulgala Bali Iena and other sites from Sri Lanka, in particular the question of presence/absence of standardised geometric microliths. These are absent from the assemblages reported in this paper, unlike at other sites from Sri Lanka such as Batadomba-Iena. How can this be explained? What could this mean?

We have now added new sentences explaining our hypothesis relating to the absence of retouched tools in the three cave sites. This reads as follows:

“In previous fieldwork at the site, only 27 backed microlithic were found [35], a number that is extremely low in comparison to the tens of thousands of quartz lithic items discovered. A similar pattern was also observed at Fa-Hien Lena [38] and Batadomba-lena [36], where the number of backed microliths is extremely limited and retouched tools absent. These data suggest that the production of retouched tools or the resharpening of quartz flakes was unnecessary for the exploitation of the tropical environment.”

4) Bone points: as the complementarity between lithic and bone tools is mentioned at several places in the manuscript and forms an important idea in the discussion, I would suggest adding a figure showing some of these artefacts.

We have now added a new Figure 2 showing the bone points.

5) Eventually, as part of the Open Data scheme of the journal, I would suggest adding tables with all the measurements of the individual artefacts and possibly some diacritic diagrams of the knapping techniques identified as Supplementary Material.

We have added an Excel file with the measurements of the individual artefacts and some diacritic diagrams of the knapping techniques to the Supplementary Information.

Reviewer #2

This is a very interesting paper on the Late Pleistocene-Holocene lithic assemblage of Kitulgala Beli-lena. The corpus of data is robust, the paper is well written and the discussion is interesting. The paper is mainly descriptive but there is not a problem with that. My apologies for my poor English writing. If there is something not clear on my comments. I will be happy to clarify via email. Moreover, I would like to apologize for sending my review one week late, I know as an author how frustrating is to be waiting to the reviewers.

We thank the reviewer for their kind words on our paper and are delighted they found it to be “interesting”, “robust” and “well written”.

I have some remarks and suggestions, mainly to strengthen this manuscript.

1) I have some concerns regarding the stratigraphical research units (divisions) analysed. The description of the stratigraphy is very short, the stratigraphy shown in Figure 1 is quite simple and the decision of merging many layers as units of study for the lithics should be explained and justify. Figure 1 shows in the right-hand size a stratigraphy. The Late Pleistocene seems to have several layers. A better description of the stratigraphy and the decision of merging the lithic analysis in three big chronological phases should be justified and properly explained. First, I would dedicate some part of the introduction on the purely description of the stratigraphy and later on, in the methodology/sample section, I would explain why the authors use such big units. Using these three big chronological groups (Late Pleistocene, Terminal Pleistocene and Holocene) might be masking some diachronic variations (which would change completely the discussion of this paper). Maybe not, but then it must be justify. Moreover, I do not know if I missed it but I would specify what is the surface excavated and the relationship of litres of sediment/artefacts.

The primary description of the stratigraphy is presented in Wedage et al. 2020. We have now added a new description of the stratigraphic sequence and, in the Methodology section, have justified our preference for undertaking the technological description of the lithic assemblage by chronological units. In order to support our interpretation of long-term technological stability at the site, we have included a new S1 Table in the Supplementary Information, showing the number of lithic artefacts by context. We have also included a new S2 Table presenting the types of core by contexts.

2) Since the beginning of the manuscript, they use the term ‘microlithism’. In the methodology section they explain that they follow Shea and Pargeter definition. That clarification is pertinent as there is a lot of confusion regarding that term in lithic analyses, but I would explain this before in the manuscript. Because when I started reading, I was wondering what definition are the authors following and I wrote it in my notes. On this topic, in the last paragraph of the Summary, there is no need to mention that there are not backed pieces.

We have now specified in the introduction that we consider microliths as lithic items smaller than 40 mm:

“Reassessment of cultural materials have demonstrated that technical behaviours were associated with the consistent production of bone points and quartz microliths (< 40 mm) [30, 33, 34, 36, 41, 48].”

3) Regarding the quartz variants, as it is the main raw material I wonder if the authors could differentiate varieties within the groups, as for example suggested by: Martínez Cortizas, A. and C. Llana Rodríguez (1996) Morphostructural variables and the analysis of their effect on quartz blank charecteristics. In: Non-Flint Stone Tools and the Palaeolithic Occupation of the Iberian Peninsula, edited by N. Moloney, L. Raposo and M. Santonja. Pp. 49-53. Tempus Reparatum, 649 Oxford. I have worked quite a bit with quartz and even if quartz is quartz for a geologist, for a stone toolmaker there are very different quality of quartzs depending if it is grainy, if it has flaws, etc which affect a lot on what you can make or not with the raw material. I am not proposing that the authors re-do the analysis, but again, I would explain the categories chosen / divisions made regarding raw material for their analysis. In addition, maybe for future analysis this is an interesting suggestion.

We have now added a description of the five categories of quartz we used during the analysis as follows: 

“We divide the quartz pebbles into five main categories based on their petrological features [128]: crystal – a translucent automorphic quartz; milky – a grainy xenomorphic quartz characterized by chalky/cloudy colour tonalities; rose – a grainy xenomorphic quartz characterized by light pink colour tonality; vein – a grainy xenomorphic quartz characterized by few reddish linear inclusions; granular – a course-grained xenomorphic quartz. ” 

4) “We first tested the normality of the data using Shapiro- Wilk tests, and, then, we employed a non-parametric test” I suppose this is because all the data were not normal distributions, but where is the table showing these results? This should be provide somewhere in the paper (Supplementary material or in a table). In this regard, I wonder if the authors are sharing their whole database. When I have published in PlosOne I am always encouraged to share my data and I think the authors should do it.

In the Supplementary Information, we have now added a new S3 Table with the p values of the Shapiro-Wilk normality tests and an Excel file with the measurements of individual artefacts.

5) “Following the demonstration of the bimodal distribution of blade sizes in southern India [113], we employ a 40mm size threshold and describe flakes, blades (bladelets), and retouched tools smaller than 40mm as microlithic (see [38])”. I suppose it is okay to compare with other data, but I would rather show the distribution in length and width of the different blanks (maybe with simple histograms) and asses if the distribution is continuous or there are ‘breaks’ on it. This seems particularly relevant being so homogeneous all the groups selected.

In Figure 7, we have now presented a histogram of the frequency of complete flakes by length intervals during the different chronological phases at Kitulgala Beli-lena showing no distribution breaks. In the Supplementary Information, we have added a new S1 Figure presenting a histogram of the frequency of complete flakes by width intervals.

6) The anvil rested modality I think it was firstly defined by Callahan. I think it should be referenced (the work I am referring is a special monograph on the Neolithic of Sweden, If I remember correctly) In one of their groups they recognize this type of knapping, it would be convenient to specify how they recognize this, what are the stigma? Also, I would add some photos, etc I have been working on bipolar knapping for a while and I do not think the identification of this type of knapping/variant is straightforward, there is very little experimentation. I ask because I am curious and I am working on it indeed.

We have added the citation of Callahan 1987 and explained in more detail how we interpreted the core. We have also provided a new S2 Figure showing the core reduced by using the anvil rested modality.

7) A figure explaining the different types of cores is needed the variants vertical, horizontal, bipolar+unid., unidirectional, etc. In lithic technology every analyst use slightly different categories and for the shake of clarity and future comparisons. I recommend this is better explained. In the same vein, what do the authors mean by ‘core-edge’ flakes? Also, regarding ‘debris’ I think I have not seen an explanation of how the authors define this category (well, where they put the measure cut to consider a flake chips or debris).

We have provided a new Figure 3 showing the categories of different types of cores used in the analysis. 

We have explained, in the text of the results, that the category “bipolar + unidirectional” referrers to exhausted bipolar cores that were knapped for a short sequence by free hand using the simple unidirectional modality. Conversely, the category “unidirectional” referrers to cores/chunks knapped using the simple unidirectional modality. 

We have corrected the term “core-edge flakes” by changing it to “core-edge removal flakes”. We consider these flakes, blanks that retain a small portion of the core of their lateral edge. In the Methods section, we state that we consider “debris” (now chips) to be all the fragments smaller than 10mm.

8) I also have a problem with the use of the term ’splintered pieces’, particularly in an assemblage with a lot of bipolar knapping. It seems the authors are referring to a type of blank (which is not its original definition which was typological). The authors need to give clarity in this regard too. Splitered pieces or pieces esquillees was originally a typological category defined by Sonneville-Bordes, later on was problematic, thus they eliminated it from the type-list of the Upper Palaeolithic because it was considered an a posteiori tool…It is problematic since long time ago but researchers keep using it I do not know why. I would dedicate some explanation too to the different blanks considered in this analysis from bipolar knapping: what is a flake-splinter? How is that category different to what they call splintered flake? I would show all this in methodology, particularly because I Imagine in the future they will do a comparative analysis with other sites in Sri Lanka and it is important to explain all these categories better. I cannot remember if these are the categories used in : Mourre V. Les Industries en Quartz au Paléolithique. Terminologie, Méthodologie et Technologie. Paléo. 1996;8:205-23. Regarding this publication I think to reference it is interesting but the authors must bear in mind that the type of knapping defined by Mourre in that work is to flake big blanks, whereas the type of bipolar knapping they have is towards the production of small things, which is slightly different. To distinguish variants within bipolar knapping is the future as finally researchers are opened to identify it.

We do not consider “splinter flakes” as retouched tools as described in the typological work of Sonneville-Bordes. In the Methods section, we state that we interpret “splinter flakes” as “splintered pieces with more or less pronounced traces of longitudinal fracture”. We used the term “splinter/ splintered” for evidencing that the blank retains small portions of the core also in the distal side of the flake. We have now elaborated on these criteria in the text and we have corrected the name of this category in Tables 1, 6, and 9.

Regarding the work of Mourre 1996, we agree with the reviewer that this methodology was developed originally to analyze larger flakes. However, in our assemblage, we have documented several examples of siret breakages that are in accordance with those described in that publication and, for this reason, we choose to follow it.

9) Regarding the comparison of the three groups of lithics analysed I would try to do some comparative statistical test beyond figure 5 and 9, although these two are quite eloquent.

We have provided information on the statistical analysis that are significant for comparing the three assemblages in the main text.

10) Finally, how did they differ between crystal and milky bipolar cores when the cores were very reduced/exhausted? By the presence of facets? Translucency? I am curious as I have this problem often.

We have differentiated crystal quartz cores from milky quartz based on macroscopic observations of the raw materials. Crystal quartz cores are generally transparent and translucent whereas milky quartz artefacts remain opaque and chalky also when the dimension are reduced. 

11) Finally, I would like to say that I enjoyed the reading a learning from this fascinating assemblage.

We thank the reviewer for the kind words. We also appreciate the considered and detailed recommendations of the reviewer, which has improved our manuscript.

---

## [Decision Letter · Decision Letter 1]

9 Aug 2022

Homo sapiens lithic technology and microlithization in the South Asian rainforest at Kitulgala Beli-lena (c. 45 – 8,000 years ago)

PONE-D-22-05404R1

Dear Dr. Picin,

We’re pleased to inform you that your manuscript has been judged scientifically suitable for publication and will be formally accepted for publication once it meets all outstanding technical requirements.

Kind regards,

Raven Garvey, Ph.D.

Academic Editor

PLOS ONE

Additional Editor Comments (optional):

Reviewers' comments:

Reviewer's Responses to Questions

**Comments to the Author**

1. If the authors have adequately addressed your comments raised in a previous round of review and you feel that this manuscript is now acceptable for publication, you may indicate that here to bypass the “Comments to the Author” section, enter your conflict of interest statement in the “Confidential to Editor” section, and submit your "Accept" recommendation.

Reviewer #1: All comments have been addressed

Reviewer #2: All comments have been addressed

2. Is the manuscript technically sound, and do the data support the conclusions?

Reviewer #1: Yes

Reviewer #2: Yes

3. Has the statistical analysis been performed appropriately and rigorously? 

Reviewer #1: Yes

Reviewer #2: Yes

4. Have the authors made all data underlying the findings in their manuscript fully available?

Reviewer #1: Yes

Reviewer #2: No

5. Is the manuscript presented in an intelligible fashion and written in standard English?

Reviewer #1: Yes

Reviewer #2: Yes

6. Review Comments to the Author

Reviewer #1: (No Response)

Reviewer #2: All the comments and suggestions that I pointed out have been addressed. As I said in my previous review I have enjoyed and learned from this work.

For the question: "Have the authors made all data underlying the findings in their manuscript fully available?'' I have answer no, because I do not think the attribute analysis/database has been included in the manuscript as SM. I think this is desirable. The databases should be share for future analyses, archaeological comparissons, etc. When I have published in this journal the editors have encouraged me to do so and I think the authors should do the same in this case. Lithic analyses should be as more transparent and clear as possible.

7. PLOS authors have the option to publish the peer review history of their article (what does this mean?). If published, this will include your full peer review and any attached files.

Reviewer #1: No

Reviewer #2: **Yes: **Paloma de la Peña

---

## [Editor Report · Acceptance letter]

13 Sep 2022

PONE-D-22-05404R1 

*Homo sapiens* lithic technology and microlithization in the South Asian rainforest at Kitulgala Beli-lena (*c*. 45 – 8,000 years ago) 

Dear Dr. Picin:

I'm pleased to inform you that your manuscript has been deemed suitable for publication in PLOS ONE. Congratulations! Your manuscript is now with our production department. 

Kind regards, 

on behalf of

Dr Raven Garvey 

Academic Editor

PLOS ONE